

# A Salinity Module for SWAT to Simulate Salt Ion Fate and Transport
# at the Watershed Scale
Ryan T. Bailey[1*], Saman Tavakoli-Kivi[1], Xiaolu Wei[1]
[1] Department of Civil and Environmental Engineering, Colorado State University, 1372 Campus Delivery, Fort Collins, CO,
80523-1372, United States.
*Correspondence to*: Ryan Bailey (rtbailey@colostate.edu)
**Abstract.** Salinity is one of the most common water quality threats in river basins and irrigated regions worldwide. However, no
available numerical models simulate all major processes affecting salt ion fate and transport at the watershed scale. This study
presents a new salinity module for the SWAT model that simulates the fate and transport of 8 major salt ions ($SO_4$, $Ca$, $Mg$, $Na$,
$K$, $Cl$, $CO_3$, $HCO_3$) in a watershed system. The module accounts for salt transport in surface runoff, soil percolation, lateral flow,
groundwater, and streams, and equilibrium chemistry reactions in soil layers and the aquifer. The module consists of several new
subroutines that are imbedded within the SWAT modelling code and one input file containing soil salinity and aquifer salinity
data for the watershed. The model is applied to a 732 $km^2$ salinity-impaired irrigated region within the Arkansas River Valley in
southeastern Colorado, and tested against root zone soil salinity, groundwater salt ion concentration, groundwater salt loadings to
the river network, and in-stream salt ion concentration. The model can be a useful tool in simulating baseline salinity transport
and investigating salinity best management practices in watersheds of varying spatial scales worldwide.
**1 Introduction**
Salinity is one of the most common water quality threats in river basins and irrigated regions worldwide. Sustainability of
crop production in irrigated areas in semi-arid and arid areas is threatened by over-irrigation, poor quality of irrigation water
(high salinity), inadequate drainage, shallow saline groundwater, and salinization of soil and underlying groundwater, all of
which can lead to decreasing crop yield. Of the estimated 260 million ha of irrigated land worldwide, approximately 20-30
million ha (7-12%) is salinized (Tanji and Kielen, 2002), with a loss of 0.25 to 0.5 million ha each year globally. Approximately
8.8 million ha in western Australia alone may be lost to production by the year 2050 (NLWRA, 2001), and 25% of the Indus
River basin is affected by high salinity. Within the western United States, 27-28% of irrigated land has experienced sharp
declines in crop productivity due to high salinity (Umali, 1993; Tanji and Kielen, 2002), thereby rendering irrigated-induced
salinity as the principal water quality problem in the semi-arid regions of the western United States.
Salinization of soil and groundwater systems is caused by both natural processes and human-made activities. Salt naturally
can be dissolved from parent rock and soil material, with salt minerals (e.g. gypsum $CaSO_4$, halite $NaCl$) dissolving to mobile
ions such as $Ca^{2+}$, $SO_4^-$, $Na^+$, and $Cl^-$. In addition, salt ions can accumulate in the shallow soil zone due to waterlogging, which is
a result of over-irrigating and irrigating in areas with inadequate drainage. Salts moving up into the soil zone can become evapo-
concentrated due to the removal of pure water by crop roots. Soil water salinization leads to a decrease in osmotic potential, i.e.
the potential for water to move from soil to the crop root cells via osmosis, leading to a decrease in crop production.
Numerical models have been used extensively to assess saline conditions, simulate salt movement across landscapes and
within soil profiles, predict salt build-up and movement in the root zone, and investigate the impact of best management
practices (Oosterbaan, 2005; Schoups et al., 2005; Burkhalter and Gates, 2006; Singh and Panda, 2012). Available models that
either have inherent salinity modules or can be applied to salinity transport problems include UNSATCHEM (Šimůnek and





Suarez, 1994), HYDRUS linked with UNSATCHEM (Šimůnek et al., 2012); DRAINMOD, LEACHC (Wagenet and Hutson,
1987), SAHYSMOD (Oosterbaan, 2005; Singh and Panda, 2012), CATSALT, and MT3DMS (Burkhalter and Gates, 2006).
Whereas several of these models include major ion chemistry for salt ions (e.g. precipitation-dissolution, cation exchange,
complexation) (UNSATCHEM, HYDRUS), their application typically is limited to small field-scale or soil-profile domains (e.g.
Kaledhonkar and Keshari, 2006; Schoups et al., 2006; Kaledhonkar et al., 2012; Rasouli et al., 2013). Conversely, models such
as SAHYSMOD and MT3DMS have been applied to regional-scale problems, but lack the reaction chemistry and treat salinity
as a conservative solute. SAHYSMOD uses seasonal water and salt balance components for large-scale systems on a seasonal
time step (Singh and Panda, 2012). MT3DMS is a finite-difference contaminant transport groundwater model that uses
MODFLOW output for groundwater flow rates, but does not include salt ion solution chemistry (Burkhalter and Gates, 2006).
Schoups et al. (2005) used a hydro-salinity model that couples MODHMS with UNSATCHEM to simulate subsurface salt
transport and storage in a 1,400 km$^2$ region of the San Joaquin Valley, California. The model, however, does not consider
salinity transport in surface runoff or salt transport in streams, limiting results to soil salinity and groundwater. Currently, there is
no model that simulates salt transport in all major hydrologic pathways (surface runoff, soil percolation and leaching,
groundwater flow, streamflow) at the watershed-scale that also considers important solution reaction chemistry. Such a model is
important for assessing watershed-scale and basin-scale salt movement and investigating the impact of large-scale salinity
remediation schemes.
The objective of this paper is to present a salinity transport modeling code that can be used to simulate the fate and transport
of the major ions (SO$_4$, Ca, Mg, Na, K, Cl, CO$_3$, HCO$_3$) in a watershed hydrologic system. The salinity module is implemented
within the SWAT modeling code, and thereby salt transport pathways include surface runoff, percolation, soil later flow,
groundwater flow and streamflow. The soil water and groundwater concentration of each salt ion is also affected by equilibrium
chemistry reactions: precipitation-dissolution, complexation, and cation exchange. The use of the model is demonstrated through
application to a 732 km$^2$ region of the Lower Arkansas River Valley (LARV) in southeastern Colorado, an irrigated alluvial
valley in which soil and groundwater salinization has occurred over the past few decades. The model is tested against salt ion and
total dissolved solids (TDS) concentration in surface water (Arkansas River and its tributaries), groundwater (from a network of
monitoring wells), and soil water (from a large dataset of soil salinity measurements). The salinity module for SWAT can be
applied to any watershed to simulate baseline conditions and to test the effect of best management practices on watershed
salinity.

**2 Development of the SWAT Salinity Module**
This section provides a brief overview of the SWAT model, followed by a description of the SWAT salinity module. Sect. 3
demonstrates the use of the salinity module to a regional-scale irrigated stream-aquifer system in the Lowe Arkansas River
Valley, Colorado.
**2.1 The SWAT Model**
The SWAT (Soil and Water Assessment Tool, Arnold et al., 1998) hydrologic model simulates water flow, nutrient
mass transport and sediment mass transport at the watershed scale. It is a continuous, daily time-step, basin-scale, distributed-
parameter watershed model that simulates water flow and nutrient (nitrogen, phosphorus) transport in surface runoff, soil
percolation, soil later flow, groundwater flow and discharge to streams, and streamflow. The watershed is divided into subbasins,
which are then further divided into multiple unique combinations (Hydrologic Response Units HRUs) of land use, soil type and
topographic slope for which detailed water and nutrient mass balance calculations are performed. Routing algorithms route water
and nutrient mass through the stream network to the watershed outlet. SWAT has been applied to hundreds of watersheds and



river basins worldwide to assess water supply and nutrient contamination under baseline conditions (Abbaspour et al., 2015) and
scenarios of land use change (Zhao et al., 2016; Zuo et al., 2016; Napoli et al., 2017), best management practices (Arabi et al.,
2006; Maringanti et al., 2009; Ullrich and Volk, 2009; Dechmi and Skhiri, 2013), and climate change (Jyrkama and Sykes, 2007;
Ficklin et al., 2009; Tweed et al., 2009; Haddeland et al., 2010; Brown et al., 2015). However, it has not yet been applied to
salinity issues.
**2.2 Salinity Module for SWAT**
The new SWAT salinity module simulates the fate and transport of 8 major salt ions (SO₄, Ca, Mg, Na, K, Cl, CO₃, HCO₃)
via surface runoff, soil later flow, soil percolation and leaching, groundwater flow, and streamflow, subject to chemical reactions
such as precipitation-dissolution, complexation, and cation exchange within soil layers and the alluvial aquifer. The module also
simulates the loading of salt mass to the soil profile via saline irrigation water from both surface water (subbasin channel) and
groundwater (aquifer) sources. A watershed cross-section schematic describing these processes is shown in Figure 1.
The salinity module is implemented directly into the SWAT FORTRAN code, with new subroutines developed for salt
chemistry (*salt_chem*), salt irrigation loading (*salt_irrig*), salinity percolation and leaching (*salt_lch*), and salt groundwater
transport and loading to streams (salt_gw). Other standard SWAT subroutines are modified to incorporate salt ion transport and
effects, such as SWAT's crop growth modules, lagging solutes in surface runoff and groundwater flow (*surfstor*, *substor*), and
routing solutes through the stream network (*watqual*). These subroutines are shown in Figure 2 within the general SWAT
modeling code data flow. For each day loop, the mass balance calculations for each HRU are performed. Salt subroutines are
shown for chemical equilibrium, irrigation loading, salt leaching, soil salinity stress, salt groundwater transport and loading, and
lagging in surface runoff and groundwater flow. At the end of the HRU calculations, the water, sediment, nutrients, and salt mass
is routed through the stream network, with in-stream concentration of each salt ion simulated for each SWAT subbasin. Details
for each salt ion process are now presented. For the equations presented, *S* refers to salt mass, and the subscript *i* refers to the 8
major ions. For the transport equations, calculations are similar to SWAT's transport equations for nitrate. Salinity module input
data and output data also will be discussed later in this section.
**2.2.1 Salt in Surface Runoff ("salt_lch" and "surfstor" subroutines)**
The mass of each salt ion can be transferred from an HRU to the subbasin channel via surface runoff. The salt ion mass
generated in surface runoff $S'_{i,surf}$ (kg/ha) for the current day is calculated as:
$$S'_{i,surf} = \beta_{S_i} \cdot C_{S_i} \cdot Q_{surf} \quad (1)$$
where $\beta_{S_i}$ is the salinity percolation coefficient, $C_{S_i}$ is the concentration of the $i^{th}$ salt ion in the mobile water for the top 10 mm
of soil (kg salt /mm water), and $Q_{surf}$ is the surface water generated from the HRU on a given day (mm water). As only a portion
of the surface runoff and lateral flow reaches the subbasin channel on the day it is generated, SWAT uses a storage feature to
surface runoff. The salt ion mass reaching the subbasin channel on the current day via surface runoff is calculated as:
$$S_{i,surf} = \left(S'_{i,surf} + S_{i,surfstor}\right) \cdot \left(1 - \exp\left[\frac{-surlag}{t_{conc}}\right]\right) \quad (2)$$
where $S_{i,surf}$ is the mass of the $i^{th}$ salt ion that reaches the subbasin channel on the current day (kg/ha), $S_{i,surfstor}$ is the salt ion
surface runoff stored or lagged from the previous day (kg/ha), *surlag* is the surface runoff lag coefficient, and $t_{conc}$ is the time of
concentration for the HRU (hrs).
**2.2.2 Lateral Flow ("salt_lch" and "substor" subroutines)**
The salt ion mass generated in lateral flow $S'_{i,lat,ly}$ (kg/ha) from a soil layer for the current day is calculated as:





$S'_{i,lat,ly} = C_{S_i} \cdot Q_{lat,ly}$    (3)
where $Q_{lat,ly}$ is the water discharge from the layer by lateral flow (mm water). Similar to surface runoff, only a portion of the
lateral flow will reach the subbasin channel on the day it is generated, and thus the salt ion mass reaching the channel on the
current day $S_{i,lat,ly}$ (kg/ha) via lateral flow is calculated as:
$S_{i,lat,ly} = \left( S'_{i,lat,ly} + S_{i,latstor} \right) \cdot \left( 1 - \exp\left[ \dfrac{-1}{TT_{lat}} \right] \right)$    (4)
where $S_{i,latstor}$ is the salt ion mass stored or lagged from the previous day (kg/ha) and $TT_{lag}$ is the lateral flow travel time (days).
**2.2.3 Soil Percolation ("salt_lch" subroutine)**
The salinity module tracks the mass of each salt ion (kg/ha) in each soil layer. The salt ion mass moved to the underlying
soil layer by percolation $S_{i,perc,ly}$ (kg/ha) is calculated as:
$S_{i,perc,ly} = C_{S_i} \cdot Q_{perc,ly}$    (5)
where $Q_{lat,ly}$ is the amount of water percolating to the underlying soil layer on a given day (mm water). After percolation has been
simulated, the concentration of each salt ion (mg/L) in each soil layer is calculated using the area (m$^2$) of the HRU and the
volume of water in the soil layer (m$^3$). The leached salt ion mass is added to the shallow aquifer using the following:
$S_{i,rech} = \left[ \left( 1 - gw_{delay} \right) \cdot S_{i,perc} \right] + \left( gw_{delay} \cdot S_{i,rech,t-1} \right)$    (6)
where $S_{i,rech}$ is the salt ion mass loaded to the water table via recharge (kg/ha), $S_{i,perc}$ is the salt ion mass percolated from the
bottom layer of the soil profile, $S_{i,rech,t-1}$ is the leached salt ion mas from the previous day, and $gw_{delay}$ is the groundwater delay
time, i.e. the time required for water leaving the bottom of the root zone to reach the water table (days).
**2.2.4 Groundwater Flow ("salt_gw" subroutine)**
The salinity module tracks the mass of each salt ion (kg/ha) in the aquifer. The salt ion mass generated in groundwater flow
$S'_{i,gw}$ (kg/ha) from the aquifer for the current day is calculated as:
$S'_{i,gw} = C_{S_{i,gw}} \cdot Q_{gw}$    (7)
where $C_{S_{i,gw}}$ is the salt ion concentration in the aquifer (kg salt /mm water), and $Q_{gw}$ is the groundwater flow generated for the HRU
for the current day (mm water). The concentration of each salt ion in each HRU aquifer is calculated on each day by dividing the
total mass of the salt ion (g) by the total volume of groundwater (m$^3$).
**2.2.5 Streamflow ("watqual" subroutine)**
Water is routed through the watershed channel network using the variable storage routing method, a variation of the
kinematic wave model (Neitsch et al., 2011). The mass of each salt ion is routed through the channel network with water, with no
chemical reactions changing in-stream salt ion concentration. Similar to any constituent in SWAT, salt ion loadings (kg/day) can
be specified for any subbasin reach of the watershed.
**2.2.6 Salt Loading in Irrigation water ("salt_irrig" subroutine)**
Salt ion mass is added to the soil profile via irrigation water, with water derived from either the aquifer (groundwater
pumping) or from surface water diversions. Including constituent mass in irrigation water is a new feature for SWAT, as the
original code does not account for nutrient (N, P) mass in irrigation water. If the irrigation water source is a subbasin reach
(surface water irrigation), the concentration of each salt ion is multiplied by the volume of applied irrigation water (depth of
water * HRU area) to determine the mass of each salt ion (kg/ha) to add to the first soil layer. If the irrigation water source is the



shallow aquifer, the concentration of each salt ion in the HRU aquifer is used to estimate salt loading to the first soil layer. The
salt ion mass is then removed from the HRU aquifer.
**2.2.7 Salt Solution Chemistry**
The salinity chemistry implemented into SWAT is based on the Salinity Equilibrium Chemistry (SEC) module developed
for soil-aquifer systems (Tavakoli-Kivi, 2018). The equations for salinity solution chemistry presented here are performed for
each HRU soil layer and for each HRU. The solution chemistry in this module is similar to that implemented in other water
chemistry models [UNSATCHEM: Šimůnek et al. (2012), PHREEQC: Parkhurst and Appelo (2013), MINTEQA2: Paz-Garcia
et al. (2013)]. Thus, only basic details are presented here.
The SEC module includes 8 aqueous components, 10 complexed species, five solid (salt mineral) species, and four exchange
species (Table 1). The 8 aqueous components ($SO_4$, Ca, Mg, Na, K, Cl, $CO_3$, $HCO_3$) are included due to their presence in the
majority of soil-aquifer systems. The five salt minerals ($CaSO_4$, $CaCO_3$, $MgCO_3$, NaCl, $MgSO_4$) also are included due to their
presence in many soil-aquifer systems, although the module can be amended to include any mineral species. The module
simulates the dissolved concentration (mg/L) of the 8 ions in soil water and groundwater and the solid mass concentration of the
five salt mineral species in the soil and the aquifer sediment according to precipitation-dissolution, complexation, and cation
exchange reactions.
For these calculations, the duration of the model time step (daily time step for SWAT) is assumed long enough for all
constituent reactions to achieve equilibrium. The concentration of species at equilibrium is calculated using a stoichiometric
algorithm approach, in which mass balance and mass action equations are solved simultaneously. This method is used in other
water chemical equilibrium packages such as PHREEQC (Parkhurst and Appelo, 2013) and MINTEQA2 (Paz-Garcia et al.,

172 2013).

*Law of Mass Action*
At equilibrium, the concentration of all reactants and products are related using the equilibrium constant $K$:
$$K = \frac{(C)^c (D)^d}{(A)^a (B)^b} \quad (8)$$
where A and B are reactants, C and D are reactants, $a$, $b$, $c$, and $d$ are constants, and the parentheses denote solute activities. The
activity of the $i^{th}$ solute, $i_A$, is computed by multiplying the activity coefficient $\gamma_i$ by the molal concentration, where $\gamma_i$ depends on
the ionic strength $I$ of the solution:
$$I = \frac{1}{2} \sum m_i . z_i^2 \quad (9)$$
where $z_i$ is the charge number of the $i^{th}$ ion and mi is the molality (mol/kg $H_2$0). $\gamma_i$ is then given as:
$$\begin{cases} \log \gamma_i = -\dfrac{A_a z_i^2 \sqrt{I}}{1 + B_a a_i \sqrt{I}} & I < 0.1 \\[3mm] \log \gamma_i = -A z_i^2 \left( \dfrac{\sqrt{I}}{1 + \sqrt{I}} - 0.3 I \right) & 0.1 < I < 0.5 \end{cases} \quad (10)$$
where $A_a$ and $B_a$ are temperature dependent constants ($A_a = 0.5085$ m$^{-1}$ and $B_a = 0.3285 \times 10^{10}$ m$^{-1}$ at 25º C) and $a_i$ is a measure of
effective diameter of a hydrated ion $i$. The first equation in (10) is the Debye-Huckle equation for dilute solutions, and the second
equation is the Davis equation.
*Mass Balance Equations*
The mass of each element in the system, either in ion or complexed form, is tracked by a set of mass balance equations.
Equations for $SO_4$, Cl, Ca, and Na are:





$SO_{4_T}=[SO_4^{2-}]+[CaSO_4^0]+[MgSO_4^0]+[NaSO_4^-]+[KSO_4^-]$    (11a)
$Cl_T=[Cl^-]$    (11b)
$Ca_T=[Ca^{2+}]+[CaSO_4^0]+[CaCO_3^0]+[CaHCO_3^+]$    (11c)
$Na_T=[Na^+]+[NaSO_4^-]+[NaCO_3^-]+[NaHCO_3^0]$    (11d)
where $T$ denotes total concentration and brackets indicate species' molality. Similar equations are written for Mg, K, $CO_3$, and
$HCO_3$.
*Precipitation-Dissolution Reactions*
Salt minerals ($AB_s$) can dissolve or precipitate according to the stoichiometric reaction
$AB_s \leftrightarrow A^+_{aq}+B^-_{aq}$    (12)
The salt mineral will dissolved if the solution is under-saturated in in regards to $A^+_{aq}$ and $B^-_{aq}$ , and will precipitate if the
solution is super-saturated. Salt minerals in the SEC module include $CaSO_4$, $CaCO_3$, $MgCO_3$, $MgSO_4$, and NaCl, due to their
common occurrence in aquifers. For example:
$CaSO_4 \leftrightarrow Ca^{2+}+SO_4^{2-}$    (13)
with a solubility product constant:
$K_{sp_{CaSO_4}}=\dfrac{(Ca^{2+})(SO_4^{2-})}{(CaSO_4)}$    (14)
Within the SEC module, minerals are added to the system one at a time, with the solubility limits of each mineral used to
determine the direction of each reaction (precipitation or dissolution).
*Complexation Reactions*
Based on the law of mass action, equilibrium equations are written for all complexed species. For example, the equation for
$CaSO_4^0$ is:
$K_{CaSO_4}=\dfrac{(Ca^{2+})(SO_4^{2-})}{CaSO_4^0}$    (15)
where $K_{CaSO_4}$ is the equilibrium constant and is equal to 0.004866. Equations and equilibrium constants for the remaining 9
complexed species are shown in Supporting Material.
*Cation Exchange Reactions*
Cation exchange is calculated to determine the sorbed and released ions from sediment surfaces to the solution. The order of
replaceability is Na > K > Mg > Ca, determined by Coulomb's Law. The cation reaction as an equivalent reactions represented
by Gapon equation:
$X_{1/mM} + 1/n\ N^{n+} = X_{1/nN} + 1/m\ X^{m+}$    (16)
where $X_{1/mM}$ is exchangeable cation $M$ on the surface (meq/100), $X_{1/nN}$ is exchangeable cation $N$ on the surface (meq/100g), $M$ and
$N$ are metal cations, and $m+$ and $n+$ are the charges of cations $M$ and $N$ respectively. Using the cation exchange capacity of the
soil and a coefficient of Gapon selectivity coefficient for each reaction, concentration of each exchangeable species is
determined.






### 2.2.8 Salinity Module Input/Output

Required data for running the SWAT salinity module include: precipitation-dissolution solubility products for the five salt minerals ($CaSO_4$, $CaCO_3$, $MgCO_3$, $NaCl$, $MgSO_4$), initial concentration of salt ions in soil water and groundwater, and initial salt mineral solid concentration (% of bulk soil) in soil and aquifer sediment. Initial concentrations are required for each HRU. However, as will be shown in Sect. 3, using uniform (i.e. all HRU values are the same) concentration values yields the same result as using spatially-variable initial concentrations, if a warm-up period of several years is used in the SWAT simulation.

All input data are provided in a single input file, "*salt_input*". To turn on the salinity module, a single line has been added at the end of the *file.cio* file, with flag being read (0 or 1) to exclude/include the salinity module. If the flag is set to 1, the SWAT code will open and read the contents of the *salt_input* file.

### 3 Application of SWAT Salinity Module to an Irrigated Stream-Aquifer System

### 3.1 Study Region: Lower Arkansas River Valley, Colorado

The salinity module is tested for a 732 km$^2$ irrigated stream-aquifer system along the Arkansas River in southeastern Colorado (Figure 3A). The region consists the Arkansas River and tributaries (e.g. Timpas Creek, Crooked Arroyo, see Figure 3A) running through and over a thin (~10-15 km in width) and shallow (~10-20 m) sandy alluvial aquifer. The climate is semi-arid, requiring irrigation to supplement rainfall for crop growth. Irrigation water is derived either from the Arkansas River via a system of irrigation canals or from the aquifer via a network of ~500 pumping wells (Figure 3A). Cultivation and associated irrigation occurs March through November.

Salinization of soil, groundwater, and surface water in the region has steadily worsened since the 1970s due to increased irrigation diversions from the Arkansas River, high water tables due to excessive water applications to fields, and the existence of salt minerals, particularly gypsum ($CaSO_4$) (Konikow and Person, 1985; Goff et al., 1998; Gates et al., 2002; Gates et al., 2016). Soil salinity levels under about 70% of the area exceed threshold tolerance for crops, with the regional average of crop yield reduction from salinity and waterlogging estimated to range from 11 to 19% (Gates et al., 2002; Morway and Gates, 2012).

From sampling groundwater from a network of 82 observation wells (see Figure 3B) (sampling from June 2006 to May 2010), average salinity concentration of shallow groundwater is approximately 2,700 to 3,000 mg/L, and annual salt loading to the Arkansas River from groundwater return flows is about 500 kg per irrigated ha, per km of the river. In the 1990s, 68% of producers stated that high salinity levels are a significant concern (Fraser et al., 1999). For the region modeled in this study, average TDS concentration ($C_{TDS}$) in groundwater is 3,334 mg/L (443 samples), with a minimum of 459 mg/L and a maximum of 44,600 mg/L. The presence of gypsum is revealed in the high concentration of SO4 ($C_{SO_4}$), with average, minimum, and maximum concentrations of 1,878 mg/L, 147 mg/L, and 29,457 mg/L, respectively. Average soil salinity, using electrical conductivity (EC), is 4.11 dS/m (54,700 measurements), with minimum and maximum of 0.9 dS/m and 56.5 dS/m, respectively. Based on 6 surface water sampling sites (4 in the Arkansas River, 2 in tributaries; Figure 3B), average $C_{TDS}$ and $C_{SO_4}$ is 1145 mg/L and 560 mg/L, respectively. More details of observed groundwater, soil water, and surface water concentrations are provided in Sect. 3.3.2 when model results are presented.

### 3.2 SWAT Model

A previously calibrated and tested SWAT model for the study region is used to simulate salt fate and transport using the developed salinity module. The SWAT model is detailed in Wei et al. (2018). The region was divided into 72 subbasins (see Figure 3B). A method was developed to apply SWAT to highly-managed irrigated watersheds, and included: designating each cultivated field as an individual HRU (see Figure 3B for the map of fields); crop rotations to simulate the effects of changing





crop types for each field during the 11-year simulation; seepage to the aquifer from the earthen irrigation canals; and SWAT's
auto-irrigation algorithms to trigger irrigation events based on plant water demand for both surface water irrigation and
groundwater irrigation. The method resulted in 5,270 HRUs. Implementing canal seepage required a slight change to the SWAT
modeling code to add pre-processed, estimated canal seepage to HRU aquifer. Canal seepage rates were obtained from field
measurements (Susfalk eta l., 2008; Martin et al., 2014). The model was run for the 1999-2009 time period, with simulated
streamflow compared to observed hydrographs at 5 stream gages (Rocky Ford, La Junta, Las Animas, Timpas Creek, Crooked
Arroyo; see Figure 3B) for model testing (Wei et al., 2018).
**3.3 SWAT Model with Salinity Module**
**3.3.1 Model Construction and Simulation**
The SWAT model is run from April 1 1999 to December 13 2009, with observed data for testing available from June 2006
to December 2009. The 1999-2005 period thus serves as a warm-up simulation period. The calibration period is 2006-2007, and
the testing period from 2008-2009. Required inputs include initial soil water and groundwater ion concentrations, initial soil and
aquifer sediment salt mineral fractions and, due to the study region being a part of the larger Lower Arkansas River Valley, ion
mass loading in the Arkansas River at the upstream end of the modeled region (Catlin Dam; see Figure 3B).
Salt ion mass loading (kg/day) in the Arkansas River at Catlin Dam were estimated using daily measured values of EC
(dS/m) and streamflow ($m^3$/s) and periodic measurements of salt ion concentration (mg/L). Linear relationships were established
between EC and the concentration of each salt ion, with this relationship then used to estimate salt ion concentration for each day
of the simulation period. The daily in-stream mass of each salt ion was then calculated by multiplying daily salt ion
concentration by streamflow, and added to the point-source SWAT input file for the appropriate subbasin. Figure 4A shows the
daily loading (kg/day) for each salt ion using this method. The make-up of total mass loading by salt ion is shown in Figure 4B,
with $SO_4$ accounting for 47% of total in-stream salt mass. The linear relationship between EC and selected salt ions ($SO_4$, Cl, Na)
and TDS is shown in the charts along the bottom of Figure 4. For TDS the $R^2$ value of the relationship is approximately 0.93.
Initial salt ion concentrations in soil water and groundwater were based on averages of observed groundwater
concentrations. For the baseline simulation, the same values were assigned to each HRU. These are 1875 mg/L, 330 mg/L, 175
mg/L, 440 mg/L, 10 mg/L, 150 mg/L, 5 mg/L, and 350 mg/L for $C_{SO_4}, C_{Ca}, C_{Mg}, C_{Na}, C_K, C_{Cl}, C_{CO_3}$, and $C_{HCO_3}$,
respectively. The effect of using spatially-varying initial concentrations is explored in additional scenarios. Salt mineral fractions
for $CaSO_4$ and $CaCO_3$ in the HRU soil layers are based on a soil survey of the region from the Natural Resources Conservation
Service (NRCS). The fraction of soil that is $CaSO_4$ and $CaCO_3$ was set to 0.1 and 0.01. For the aquifer sediment, fractions are
based on the spatial patterns determined in Tavakoli-Kivi (2018) for a salinity groundwater transport study of the same region.
Solubility products for precipitation-dissolution of salt minerals were obtained from literature and from Tavakoli-Kivi (2018)
and are 3.07 x $10^{-9}$, 4.8 x $10^{-6}$, 4.9 x $10^{-5}$, 0.0072, and 37.3 for $CaCO_3$, $MgCO_3$, $CaSO_4$, $MgSO_4$, and NaCl, respectively, for both
soil and aquifer sediments.
Only minimal manual calibration was applied to the model, to yield correct magnitudes of salt ion concentration in soil
water, groundwater, and stream water. Targeted parameters were the solubility product of $CaSO_4$ precipitation-dissolution, and
the soil fraction of $CaSO_4$. The solubility produce was increased from 0.000049 to 0.0003, and the soil fraction of $CaSO_4$ was
decreased from 0.01 to 0.009. Model results are tested against in-stream concentration of salt ions, soil water EC (dS/m),
groundwater concentration of salt ions, and groundwater salt ion mass loading to the Arkansas River. Observed soil EC values
were obtained using a saturated paste extract, and hence comparison with model results will not be as rigorous as for
groundwater and surface water data.





Several variations of the model were run to test the effect of 1) initial salt ion concentrations and 2) specified loading of salt
ion mass at the upstream end of the Arkansas River. For 1), the variations include uniform initial concentrations (baseline
model), random spatially-variable concentrations, and initial concentrations equal to 0. For 2), the variation included one
simulation with no loading.

**3.3.2 Model Results**

Model results consist of in-stream salt ion and TDS concentration, hydrologic pathway (groundwater discharge, surface
runoff, percolation) salt loadings, groundwater salt ion concentration, soil water EC, watershed-wide salt balance, and
groundwater salt loading to the Arkansas River.

**3.3.2.1 In-Stream Salt Ion Concentration**

Simulated and observed in-stream salt ion concentrations (mg/L) are shown in Figure 5 for the Rocky Ford site (Figure 5A)
and the Crooked Arroyo site (Figure 5B). Results are shown for $SO_4$, Ca, Cl, and $HCO_3$, with the calculated Nash-Sutcliffe
model efficiency coefficient (NSE) shown on each plot. Results for TDS at all 5 gaging stations are shown in Figure 6. As can be
seen by the trends in concentration and also the NSE values, the SWAT model performs well in replicating in-stream salt ion
concentrations, particularly for $SO_4$ (NSE = 0.60), Ca (NSE = 0.54), $HCO_3$ (NSE = 0.73), and TDS (NSE = 0.69) in the Arkansas
River at the Rocky Ford gaging site. The model does not perform as well in downstream sites, with NSE at La Junta and at Las
Animas equal to 0.34 and 0.25, respectively, although the trends are correct and the magnitudes are correct except for at the
downstream-most site (Las Animas), where the model under-predicts total salt concentration. This is also shown by a 1:1
comparison of all salt ion data for the Rocky Ford (Figure 7A) and Las Animas (Figure 7C) sites, which yield $R^2$ values of 0.87
and 0.74, respectively. Las Animas also has an $R^2$ value of 0.74. However, as the SWAT model often is used to estimate monthly
in-stream loads rather than daily in-stream concentration, these results are promising regarding the use of SWAT to estimate in-
stream salinity loadings.
In regards to the NSE, the model performs rather poorly in the two tributaries (Timpas Creek, Crooked Arroyo), with NSE
equal to -0.32 and 0.41, respectively, for TDS (Figure 6B, 6C). However, the overall trends and magnitude compare well to
observed data. This is shown in the 1:1 plot of all salt ion data for Timpas Creek in Figure 7B, resulting in an $R^2$ value of 0.79.
The relationship for Crooked Arroyo yields an $R^2$ value of 0.80. This is particularly promising given that there is no specified
upstream loading for the tributaries, and hence all salt mass within the stream system is due to surface runoff, lateral flow, and
groundwater discharge. Hence, comparing simulated and observed in-stream salinity concentration in these two systems is a
strong test for the model.
Figure 8 shows the salt loading via the hydrologic pathways of groundwater discharge (Figure 8A), surface runoff (8B), and
percolation from the soil profile to groundwater (8C). For Timpas Creek, 96% of salt in the creek water is from groundwater
discharge, 3% from surface runoff, and 1% from lateral flow. For Crooked Arroyo, the portions are 91%, 6%, and 3%, and for
the Arkansas River they are 96%, 3%, and 1%, highlighting the strong influence of groundwater on surface water salt load. This
is shown further by examining the domain-wide salt balance, presented in Sect. 3.3.2.3. The mass loading of total salt from the
aquifer to the Arkansas River for each day of the 2006-2009 time period is shown in Figure 9. Mass balance plot values are the
mean of a a stochastic river mass balance calculation of surface water salinity loadings along the length of the Arkansas River
within the model domain, using a method similar to Mueller-Price and Gates (2008), with values indicating the mass of salt not
accounted for by surface water loadings. These unaccounted for loadings include groundwater, and thus provide an upper limit of
in-stream salt loading from groundwater discharge.






### 3.3.2.2 Groundwater and Soil Water Salinity


Groundwater salt results are shown by spatial maps and by comparison of frequency distributions. For all simulated results,
only concentration values from days on which field samples were taken are included in the analysis. Time-averaged TDS (mg/L),
$SO_4$ (mg/L), and Na (mg/L) in groundwater is shown for each HRU in Figure 10. Also shown is soil water EC (dS/m) for each
HRU soil profile, and the percent of the soil profile (Figure 10E) and aquifer (Figure 10F) that is $CaSO_4$ (solid mineral) at the
end of the simulation period. These maps are shown to provide an indication of the degree of spatial variation simulated by the
salinity module. Variation in each system response is large, with TDS ranging from 0 to ~11,700 mg/L, $SO_4$ from 0 to ~6700
mg/L, and Na from 0 to ~1,270 mg/L. In comparison, if data from an outlier monitoring well are excluded (monitoring well with
salinity values more than double of any other monitoring well), the maximum observed values for TDS, $SO_4$, and Na are 13,000
mg/L, 6,500 mg/L, and 2,600 mg/L.
Results for all salt ions are summarized in Table 2. Average concentration of field samples (based on field samples from 82
monitoring wells shown in Figure 3B) and HRU-simulated groundwater salinity compares well, particularly for $SO_4$ (1,878 mg/L
to 2,058 mg/L) and for TDS (3,334 mg/L to 3,276 mg/L). In addition to a comparison of maximum and average values,
comparison at various magnitude levels is performed using relative frequency plots, shown in Figure 11. Results for $SO_4$ (Figure
11A), $HCO_3$ (11B), and TDS (11C) are shown. Similar to the results shown in Table 2, the comparison for $SO_4$ and TDS is good,
but the model generally under-predicts $HCO_3$ for most HRUs. A relative frequency plot of observed and simulated EC (dS/m) in
the soil profile also is shown (Figure 11D). The average of observed values and simulated values are 4.1 dS/m and 4.8 dS/m,
although the majority of observed values are between 2 dS/m and 4 dS/m whereas no such grouping occurs for the simulated
values. However, the observed data values are obtained from saturated paste extracts, which therefore lowers the salinity
concentration due to the addition of water to bring the soil to saturation. Hence, the "observed" (modified by the saturated paste
method) concentrations should be lower than what actual occurs in the field, which may explain the disagreement shown in
Figure 11D.

### 3.3.2.3 Salt Balance


The domain-wide salt balance is presented in Figure 12A. All salt balance components are included, with all values scaled
according to the small salt flux (lateral flow = 1 unit). For the soil profile, salt is added via groundwater irrigation (12 units),
surface water irrigation (33), dissolution of salt minerals (110), and upflux from the aquifer saturated zone (39), and removed via
percolation (103), surface runoff (4), and lateral flow (1). A similar salt balance can be performed for each salt ion in the system.
Salt removed from the aquifer and added to the soil profile via upflux is approximately 30% of percolation, which compares well
to a comparison of water upflux and recharge magnitudes computed by Morway et al. (2013) in a groundwater modeling study of
the region using MODFLOW.
Of the salt entering the river, 96.7% is from groundwater (151 units out of 156), and the remaining from surface runoff and
lateral flow. Time series of daily loading (kg/ha) for these three components is shown in Figure 12B, and loadings for
percolation, surface water irrigation, and groundwater irrigation are shown in Figure 12C, showing the seasonal trends in
applying irrigation water. These results also indicate that much of the salt leaching from the soil profile is due to dissolution of
salt minerals. Results also indicate the importance of including salt mass in applied irrigation water, as it accounts for
approximately half of salt leaching to the aquifer. Finally, results show the importance of including precipitation-dissolution in
the module, as this process is a large component of the salt balance. Without including this process, the module would severely
under-predict salt ion concentrations throughout the watershed, demonstrating the need to include each salt ion individually as
opposed to modeling salinity as a conservative solute in the system.






### 3.3.2.4 Scenarios and Model Guidelines


The effect of initial salt ion concentrations and upstream salt ion mas loading is summarized by the time series charts in
Figure 13. For the Rocky Ford and Las Animas gaging sites, a time series of simulated $SO_4$ (mg/L) and TDS (mg/L) is compared
for the following scenarios: uniform initial salt ion concentration ("Original": this refers to the baseline simulation); HRU-
variable initial concentration ("Variable IC"); initial concentrations equal to 0 ("Zero IC"); and not accounting for upstream salt
ion mass loading at Catlin Dam ("No US Loading").
There are only small differences between using uniform or HRU-variable initial concentrations for soil water and
groundwater. Any differences are readily resolved during the warm-up period. Hence, to facilitate model use we recommend that
uniform initial concentrations be used.
Using initial concentrations equal to 0 mg/L has a significant effect, particularly for downstream sites such as Las Animas
(Figure 13C, D). For this watershed, salt loading to the streams is principally from groundwater, and if soil water and
groundwater are not provided with initial salt ion concentrations, the groundwater salt ion loading to subbasin streams is small
compared to the baseline simulation. As downstream flow and in-stream salt loading is effected by groundwater loading, these
areas (e.g. Las Animas site) experience the effect more acutely than upstream sites such as Rocky Ford (Figure 13A,B).
However, by the end of the simulation (2009), difference between "Zero IC" and "Original" is small. This is shown by the "Diff"
time series for each plot. Therefore, if groundwater discharge is a large component of total water yield for the watershed, "Zero
IC" should not be used, or a long warm-up simulation period needs to be used.
Not including upstream salt ion loading at Catlin Dam has a stronger effect on the Rocky Ford site (Figure 13A,B) than at
the outlet (Las Animas) (Figure 13C,D). This is due to Las Animas being much farther downstream, and hence there is much
more groundwater salt ion loading to the streams that can make up for the salt not included at the upstream end of the Arkansas
River at Catlin Dam. Overall, any point sources of in-stream salt should be added, unless only downstream areas are targeted for
baseline simulations and best management practice investigation. The effect of neglecting point sources of in-stream salt
decreases as the groundwater loading component of total salt yield increases.
The importance of including equilibrium chemistry into the salt transport module is demonstrated by the results shown in
Figure 14. The simulated in-stream TDS (mg/L) is shown at the Rocky Ford site (Figure 14A), the Timpas Creek site (B), and
the Las Animas site (C), for both the original simulation (red line) and a simulation "No SEC" that does not include the SEC
module (black line). The "No SEC" simulation therefore represents a system wherein salt is transported through the stream-
aquifer system as a conservative species. Clearly, in-stream concentrations are much too low for the simulation without the SEC
module. This is due to the neglect of salt mineral dissolution, which in the actual system transfers salt mass from the soil and
aquifer material to soil water and groundwater are thereby increases the loading of salt to the stream network. For this system,
and likely most watersheds, equilibrium chemistry must be included to establish the correct magnitude of salt loading and
concentrations.

### 3.3.3 Model Use and Limitations


The salinity module of SWAT differs from other salinity models in that it accounts for salt loading for each major
hydrologic pathway in a watershed setting (stream, groundwater, lateral flow, surface runoff, tile drain flow), for each major salt
ion, subject to chemical equilibrium reactions (precipitation-dissolution, complexation, cation exchange). As such, it can be used
to estimate baseline salt loading within a watershed, and also explore the impact of land management and water management
scenarios to mitigate soil salinity, groundwater salinity, and surface water salinity. The model, however, does not simulate
physically-based, spatially-distributed groundwater flow and solute transport with an accurate depiction of water table elevation



and groundwater head gradient, and thus the trends in groundwater salt loading to streams may not be accurate (see Figure 9). To
overcome this issue, the new salinity module could be incorporated into SWAT-MODFLOW (Bailey et al., 2016), which links
SWAT and MODFLOW to simulate land surface and subsurface flow processes, and SWAT-MODFLOW-RT3D (Wei et al.,
2018), which includes reactive transport of solutes into SWAT-MODFLOW.

**4 Conclusions**

This study presents a new watershed-scale salt ion fate and transport model, by developing a salinity module for the SWAT

model. The module accounts for salt loading for each major hydrologic pathway in a watershed setting (stream, groundwater,
lateral flow, surface runoff, tile drain flow), for each major salt ion ($SO_4$, Ca, Mg, Na, K, Cl, $CO_3$, $HCO_3$). The module also
accounts for principal equilibrium chemistry reactions (precipitation-dissolution, complexation, cation exchange). For
precipitation-dissolution, five salt minerals ($CaSO_4$, $CaCO_3$, $MgCO_3$, NaCl, $MgSO_4$) have been included. The model was applied
and tested in a 732 km$^2$ irrigated stream-aquifer watershed in southeastern Colorado, along the alluvial corridor of the Arkansas
River. Model results are tested against in-stream salt ion concentration, groundwater salt ion concentration, soil salinity, and
groundwater salt loading to the Arkansas River.

The model can be used to assess baseline salinity conditions in a watershed and to explore land and water management

strategies aimed at decreasing salinization in river basins. Such strategies may include on-farm management, lining irrigation
canals to reduce saline canal seepage, dry-drainage practices, and reducing volumes of applied irrigation water. Due to the
simulation of soil water salt ion concentrations and SWAT's simulation of crop growth, the salinity module can also be used to
investigate the effect of these strategies on crop yield. Although this study applied the model to an irrigated area, the model can
be applied to non-irrigated areas as well.

**Code Availability**
The code consists of the original SWAT files, with 6 additional files for the salinity module. All files are *.f FORTRAN files.
The code is available at the following URL: https://github.com/rtbailey8/SWAT_Salinity/tree/v1.0.0 (DOI:
10.5281/zenodo.2541224). An example model input file (salt_input) and example output files are also provided.

**Author Contribution**
Ryan Bailey wrote the salinity module for SWAT and tested the module for the study region. Saman Tavakoli-Kivi prepared the
solution chemistry algorithms for the salinity module. Xiaolu Wei prepared and tested the original SWAT model for the study
region, and facilitated use of the new salinity module for the constructed SWAT model.

**Competing Interests**
The authors declare that they have no conflict of interest.

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











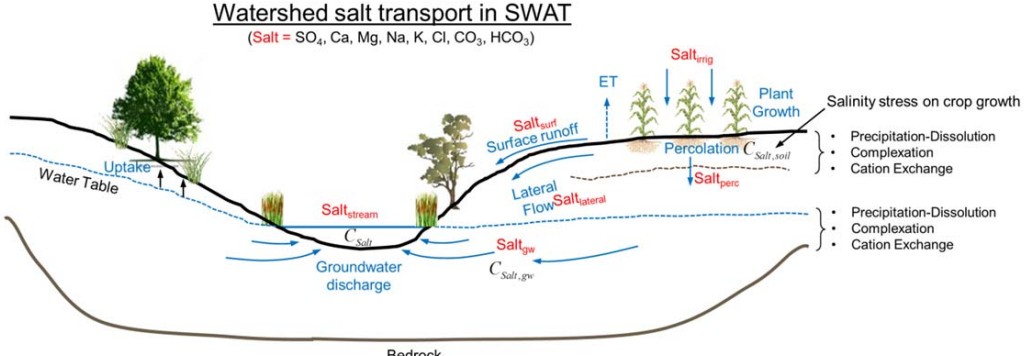

**Figure 1.** Schematic showing a cross-section of an irrigated stream-aquifer system and the major transport pathways of salt,
which consists of the eight major ions of $SO_4$, Ca, Mg, Na, K, Cl, $CO_3$, $HCO_3$. The concentration of each ion is also governed by
equilibrium chemistry reactions such as precipitation-dissolution, complexation, and cation exchange within the soil profile and
within the aquifer.

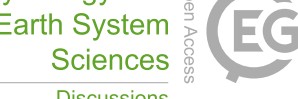

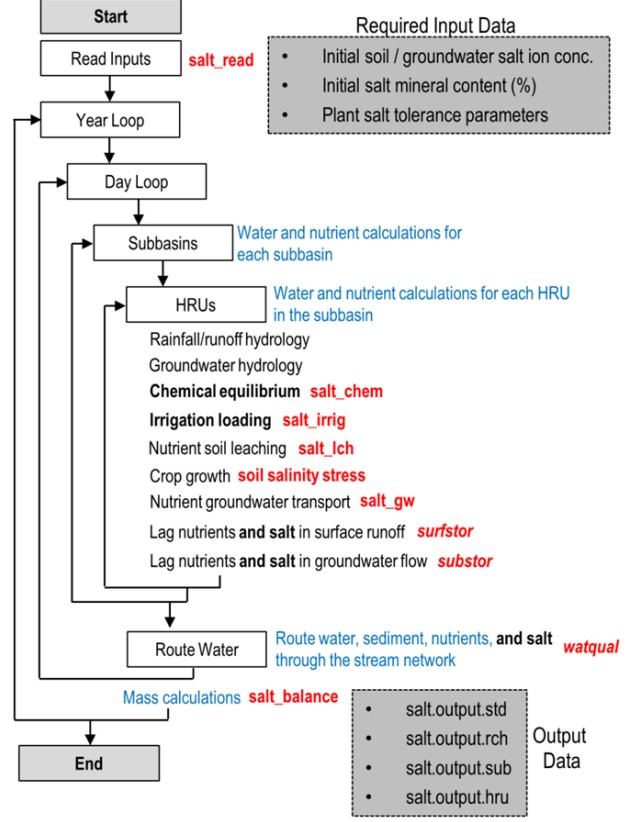

**Figure 2.** Data flow within the SWAT-Salt modeling code. Boxes and text in black and blue indicate original SWAT loops and
subroutines. Text in red indicates either new or modified subroutines for the Salinity module. The required input data for the
salinity module is shown in the upper shaded box, whereas the generated output files are shown in the lower shaded box.





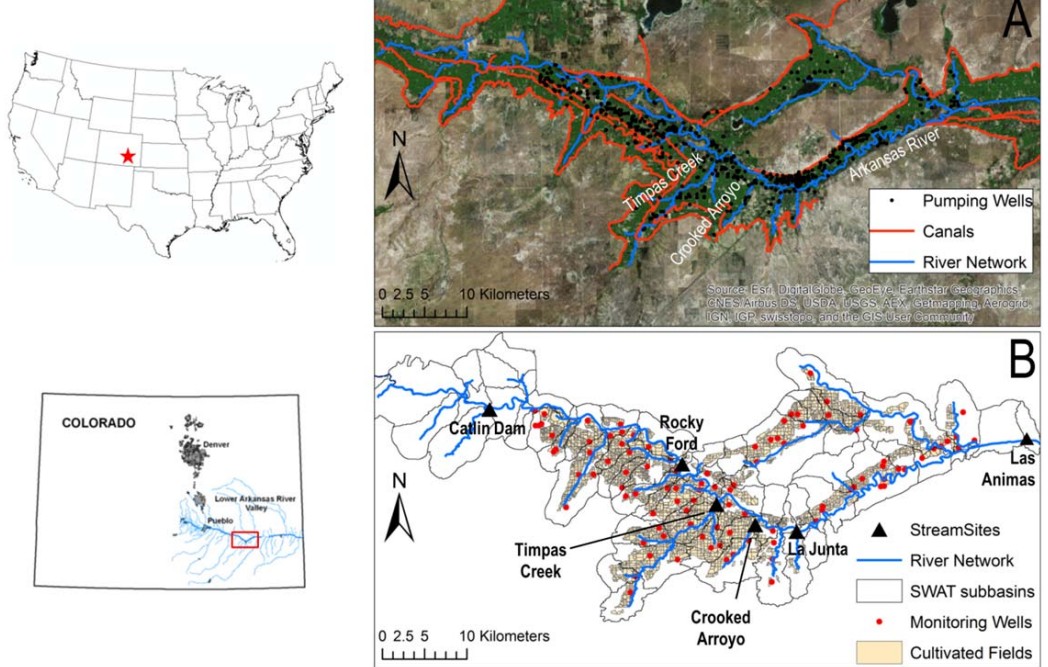

**Figure 3.** Map of study region within the Lower Arkansas River Valley of Colorado, showing (A) Arkansas River and
tributaries, irrigation canals, and pumping wells, and (B) cultivated fields, monitoring wells where groundwater is sampled for
salt ions, sampling sites where surface water is sampled for salt ions, and SWAT subbasins.




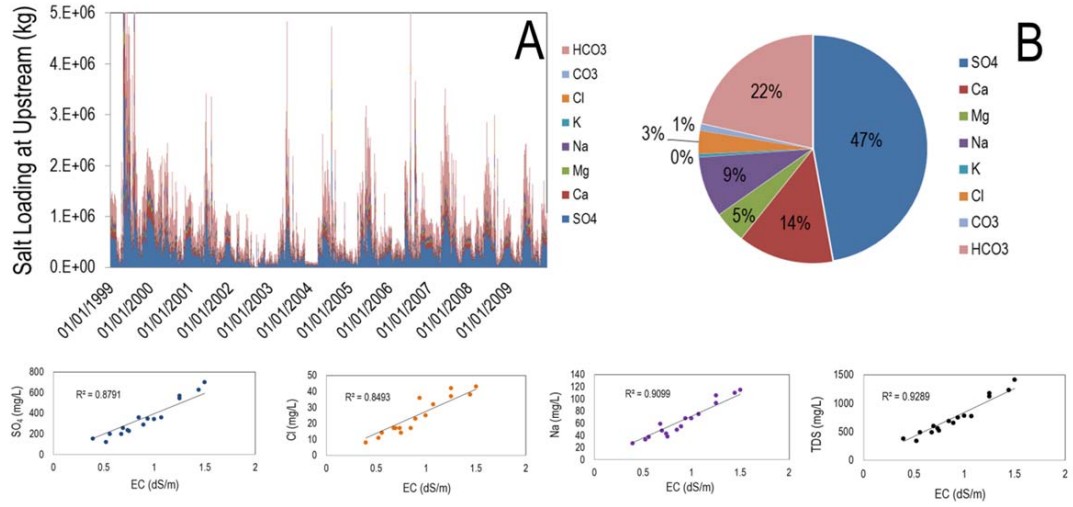

**Figure 4.** Data summarizing the specified loading of salt (kg/day) at the Catlin Dam gage site, using observed EC (dS/m) and
stream discharge (m³/day) data: (A) daily loading of salt ion, (B) percentage of total salt loading attributed to each salt ion,
(bottom charts) example regression plots used to relate EC to salt ion concentration.





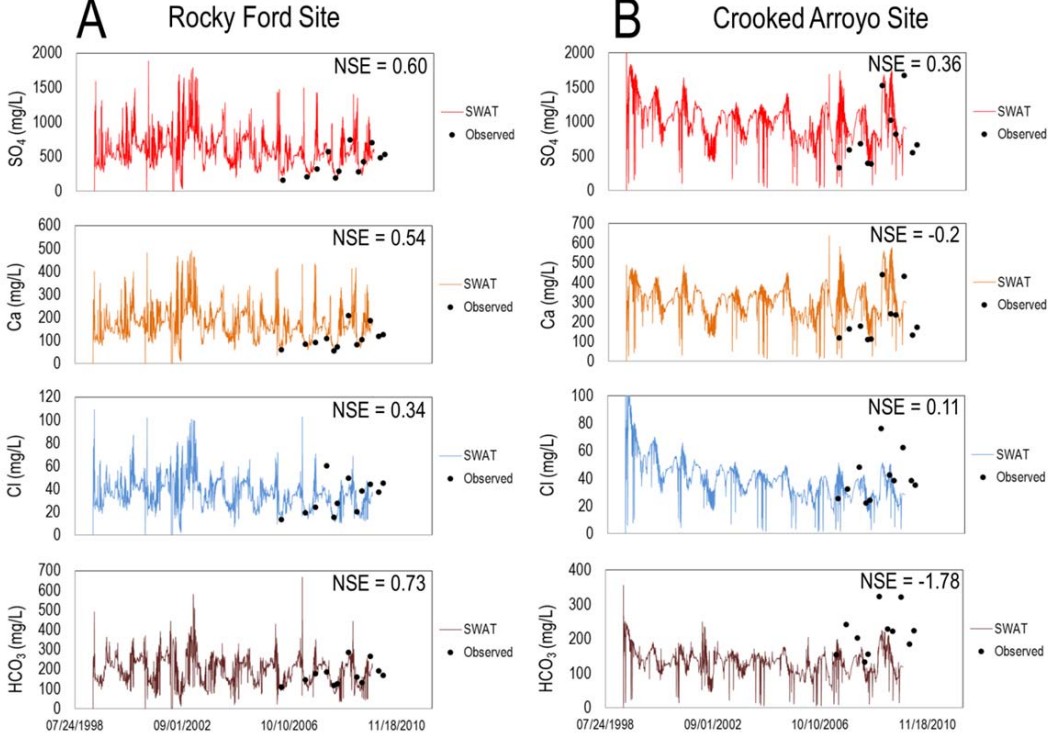

**Figure 5.** Time series of simulated and observed concentration (mg/L) of selected salt ions for the (A) Rocky Ford sampling site
along the Arkansas River (see Fig. 3) and the (B) Crooked Arroyo sampling site. The Nash-Sutcliffe model efficiency coefficient
(NSE) is shown for each plot.





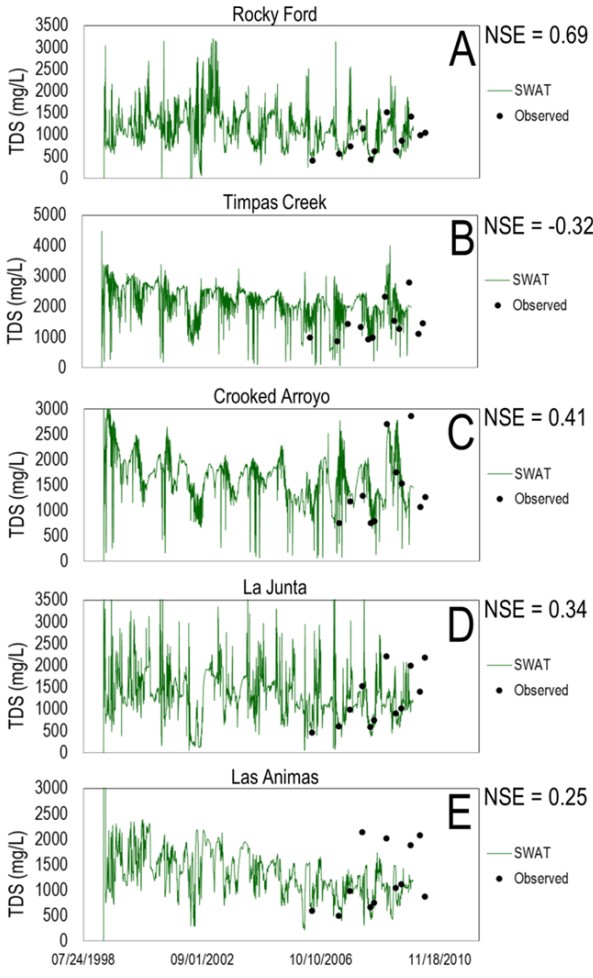

**Figure 6.** Simulated and observed total dissolved solids (TDS) (mg/L) in the five stream sampling sites along the Arkansas River
(A, D, E), and two tributaries (B, C). See Fig. 3 for locations. TDS is the summation of the concentration of the 8 salt ions. The
Nash-Sutcliffe model efficiency coefficient (NSE) is shown for each plot.





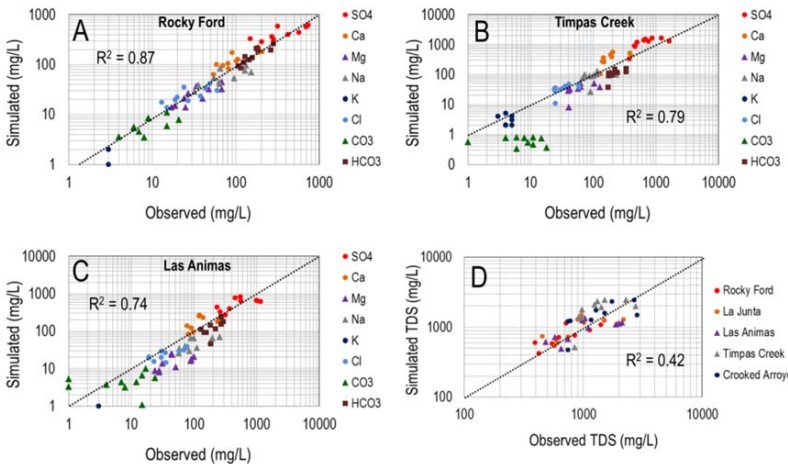

**Figure 7.** Log-log plots of observed vs. simulated salt ion concentration for the (A) Rocky Ford, (B) Timpas Creek, and (C) Las Animas surface water sampling sites. (D) shows the comparison of TDS for the five sites.

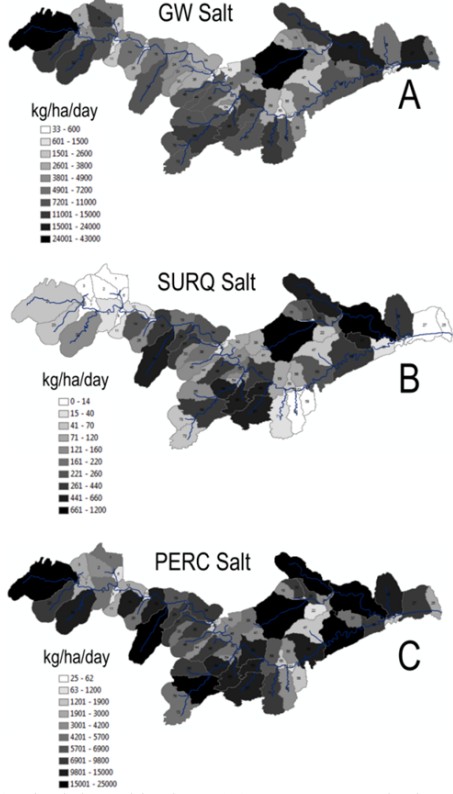

**Figure 8**. Average daily loading (kg/ha) of salt by subbasin to (A) stream network via groundwater discharge, (B) stream network via surface runoff, (C) groundwater via soil percolation.



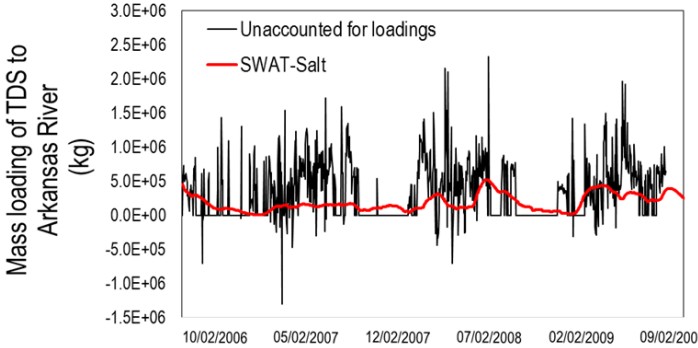

**Figure 9.** Simulated daily mass loading of TDS (kg) to the Arkansas River via groundwater discharge for the SWAT model with
uniform initial salt concentrations. Results from a salt mass balance calculation on the Arkansas River also are plotted, showing
the unaccounted for TDS loadings (groundwater, surface runoff, small inflows) in the Arkansas River.

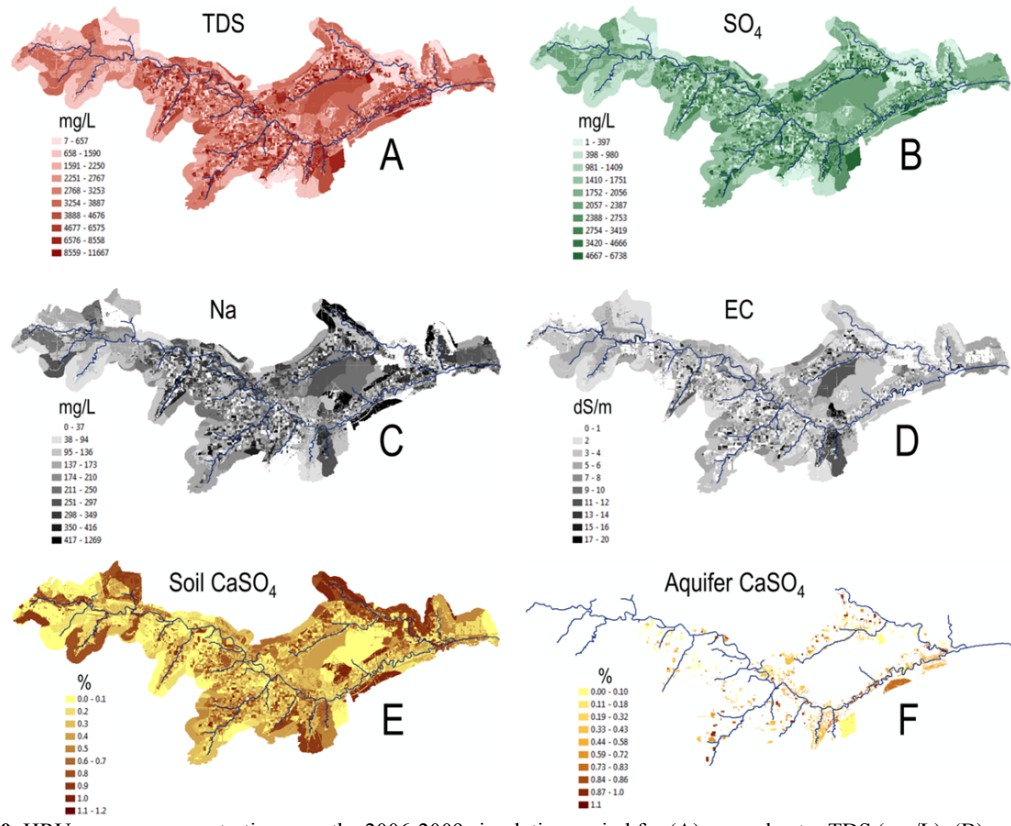

**Figure 10.** HRU average concentration over the 2006-2009 simulation period for (A) groundwater TDS (mg/L), (B) groundwater
$SO_4$ (mg/L), (C) groundwater Na (mg/L), and (D) soil water electrical conductivity EC (dS/m). (E) and (F) show percentage of
soil bulk volume and aquifer bulk volume, respectively, that is $CaSO_4$, near the end of the simulation in May 2010.






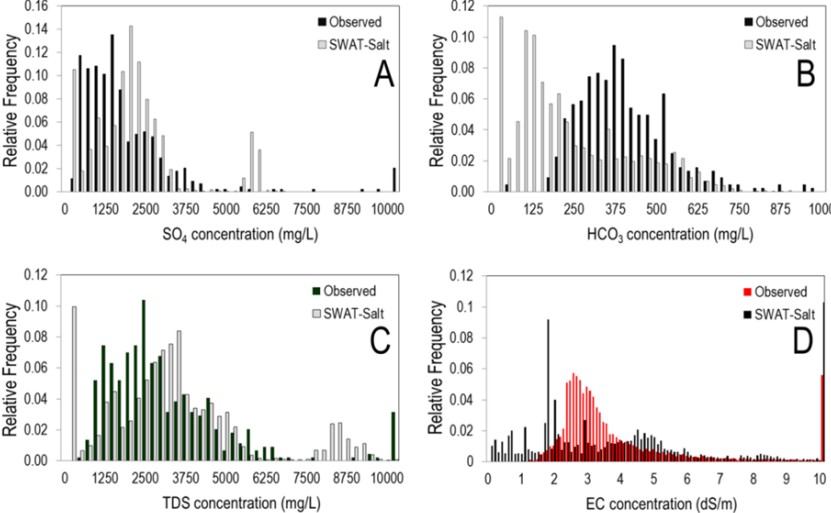

**Figure 11.** Relative frequency plots of simulated and observed values of (A) SO$_4$ groundwater concentration, (B) HCO$_3$
groundwater concentration, (C) TDS groundwater concentration, and (D) EC soil water concentration. Simulated values are
taken from each HRU of the SWAT simulation, on days for which observed values are available.

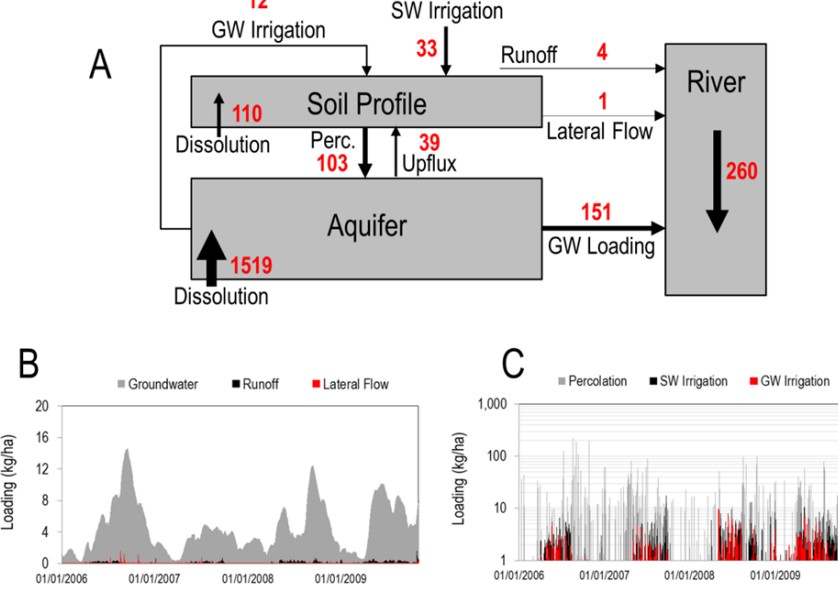

**Figure 12.** Magnitude of salt balance components in the watershed model for TDS, showing (A) relative salt flux between soil
storage compartments in the watershed for each salt transport pathway; (B) daily loading (kg/ha) of salt in groundwater, surface
runoff, and lateral flow to streams; and (C) daily loading (kg/ha) of salt in percolation water (from bottom of soil profile to the
aquifer), irrigation derived from irrigation canals, and irrigated derived from groundwater pumping.



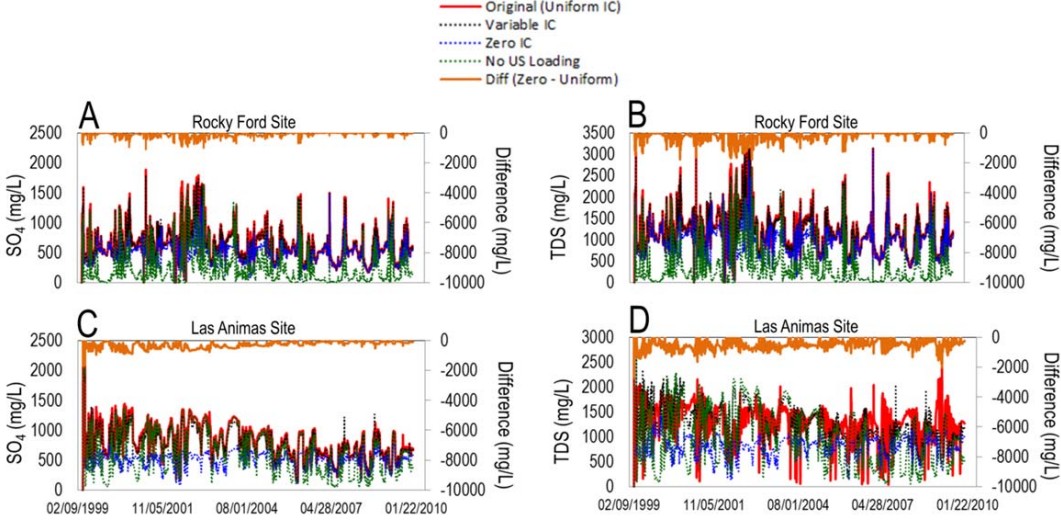

**Figure 13.** Simulated in-stream SO$_4$ and TDS concentration (mg/L) at the Rocky Ford Site and the Las Animas Site along the
Arkansas River for four scenarios: uniform initial conditions (IC) of salt soil water and groundwater concentrations,
corresponding to the original simulation; variable IC; IC = 0; and no upstream loading of salt at the Catlin Dam site. Also show
is the difference between the IC = 0 scenario and the original scenario.

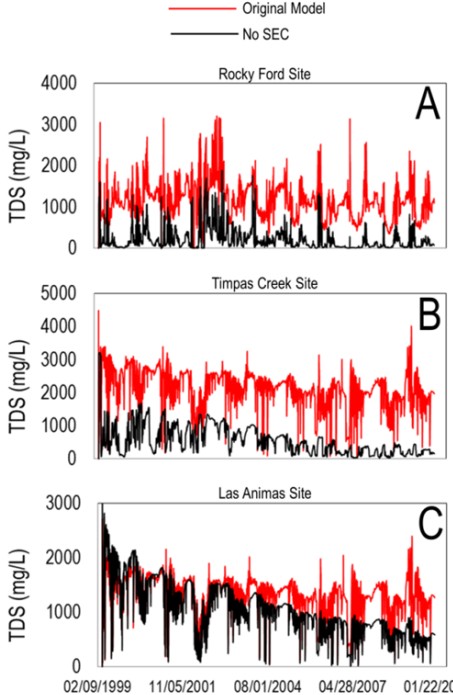

**Figure 14.** Simulated in-stream TDS concentration (mg/L) at the (A) Rocky Ford Site, (B) Timpas Creek Site, and (C) Las
Animas Site for the original simulation (red line) and a simulation without including equilibrium chemistry (SEC module) (black
line).





**Table 1.** Groups and Species included in the Salinity Equilibrium Chemistry (SEC) module for SWAT.

| Group | Species |
|---|---|
| Aqueous Species | $Ca^{2+}$, $Mg^{2+}$, $Na^+$, $K^+$, $SO_4^{-2}$, $CO_3^{2-}$, $HCO_3^-$, $Cl^-$ |
| Solid Species | CaSO4, CaCO3, MgCO3, NaCl, MgSO4 |
| Complexed Species | $CaSO_4^0$, $MgSO_4^0$, $CaCO_3^0$, $CaHCO_3^+$, $MgCO_3^0$, $MgHCO_3^+$, $NaSO_4^-$, $KSO_4^-$, $NaHCO_3^0$, $NaCO_3^0$ |
| Exchanged Species | Ca, Mg, Na, K |




**Table 2.** Summary statistics for observed (monitoring well) and simulated (SWAT) salinity concentrations in groundwater.

| Species | Maximum (mg/L) | | Average (mg/L) | |
|---|---|---|---|---|
| | Observed | Simulated | Observed | Simulated |
| Na | 2606 | 1269 | 402 | 187 |
| Ca | 767 | 2234 | 353 | 653 |
| Mg | 1019 | 497 | 191 | 78 |
| K | 85 | 277 | 4 | 9 |
| $SO_4$ | 6510 | 6738 | 1878 | 2058 |
| $CO_3$ | 42 | 8 | 2 | 0 |
| $HCO_3$ | 2362 | 1828 | 410 | 225 |
| Cl | 1803 | 480 | 95 | 65 |
| TDS | 13007 | 11667 | 3334 | 3276 |
