# Peer review of "A Salinity Module for SWAT to Simulate Salt Ion Fate and Transport 1 at the Watershed Scale 2 3"

_Hydrology and Earth System Sciences, 2018_

## Referee Comment (RC1) · Anonymous Referee #1 · 7 Mar 2019

General comments: This work focused on developing a new watershed-scale salt ion fate and transport model based on SWAT model, which can account for salt loading for each major hydrologic pathway in a watershed setting for each major salt ion (SO4, Ca, Mg, Na, K, Cl, CO3, HCO3). This is very interesting work trying to quantitatively estimate the chemical and physical characteristics of the common ions, which is important for soil salinity control in semi-arid areas with shallow water table depth. Since most current research mainly focused on the transport of total salt in surface and subsurface system while not distinguish the contribution of different ions and the reactions, this work provides the new view and method for soil salinity control. I would think this work is valuable and can be published by major revision.

Major revisions: (1) The numerical integrating method to couple the ion reactions and

water flow and solute transport model SWAT should be illustrated in details. This will help for understanding the model. (2) How many parameters were included in this model? There is no any introduction about the parameters used in the model calibration and validation, e.g., the salinity percolation coefficient $\beta Si$, the surface runoff lag coefficient surlag. How do you set the value of these parameter, which are important to judge the reasonability of the model? (3) Line 60-61," The soil water and groundwater concentration of each salt ion is also affected by equilibrium chemistry reactions: precipitation-dissolution, complexation, and cation exchange". Actually, the reactions also happen in the surface water, why not consider the chemical reactions in surface water? (4) Line 294," Only minimal manual calibration was applied to the model, to yield correct magnitudes of salt ion concentration in soil water, groundwater, and stream water. Targeted parameters were the solubility product of CaSO4 precipitation-dissolution, and the soil fraction of CaSO4." Why is only the CaSO4 used to calibrate the model? Is this due to the major ion is SO4 in this region? (5) What are the principle for setting the HRU with 5270? In Line 225, "Initial concentrations are required for each HRU." Were all the salt concentration of these 5270 HRU measured? Otherwise, how would you set the initial value? (6) Line 350. The simulations for TDS and SO4 are much better than other ions, what are the possible reasons? Is this related to the targeted parameters of CaSO4 been used in calibration mentioned in Line 294? So, if the model is used in other cases, how would you choose the targeted parameters in the calibration? How about choosing other targeted parameters in this case? (7) As shown in Fig.5, the simulation results in Rocky Ford Site are much better than those in Crooked Arroyo Site. What are the reasons? The simulation results of Na, Mg should be also shown to judge the model accuracy since the relative high concentration of these two ions as shown in Table 2. (8) From Fig.5 and Fig.6, the simulated ion concentration fluctuated much stronger than the observed value, even the simulated value closed to zero. Is this caused by the numerical instability of coupling the ion reaction module with SWAT? Or what are the major factors resulting in the strong fluctuations? (9) More discussion about the contribution of different ions on salt accumulation should be added in the

case discussion. Only the salt balance components for TDS were analyzed in Fig.12. (10) Line 329-332, are the portions of salt load calculated by the model? How would you judge the reasonability of the results?

Minor revisions: (1) Line 33, SO4-, should be SO42-. All the ions should be shown with positive and negative charges in all the other parts in the manuscript. (2) Line 88, "later", should be "lateral"? (3) Line 133, "mas", should be "mass". (4) Line 176, "C and D are reactants." Should be "C and D are products." (5) Line 177, what is the equation of iA? (6) Line 180, "mi", should be "mi". (7) Line 197, there is two "in" in the sentence. (8) Line 250, "SO4" should be "SO42-". (9) Line 295, "produce" may be "product"? (10) Line 382, "mas" should be "mass".

––––––––––––––––––––––––––––

---

## Referee Comment (RC2) · Anonymous Referee #2 · 18 Apr 2019

General comments: This work aims at simulating the fate and transport of 8 major salt ions (SO42-, Ca2+, Mg2+, Na+, K+, Cl-, CO32-, HCO3-) in a watershed hydrologic system using a new salinity transport module implemented in the SWAT code. This modelling code for salt transport includes surface runoff, percolation, soil lateral flow, groundwater flow and streamflow and also considers equilibrium chemistry reactions in soil layers and aquifers. This paper addresses with an interesting and practical approach the concerning thematic of soil and aquifer salinization. This study uses a quantification approach with salt balances performed in the watershed, includes the constituent mass in irrigation water, and the contribution of each salt ion to the salinity, which is less seen in published studies were the focus is the total of salts. Also, considering the new tool proposed that helps in predicting the impact of irrigation practices

and in controlling salinity, I suggest the publication of this work after major revision.

Specific comments: 1. Line 53 "Currently, there is no model that simulates salt transport in all major hydrologic pathways (surface runoff, soil percolation and leaching, groundwater flow, streamflow) at the watershed-scale that also considers important solution reaction chemistry." Actually there is MOHID LAND model that is also coupled with SWAT. MOHID LAND is a physically-based, spatially distributed, continuous, variable time step model for the water and property cycles in inland waters and main mediums that also includes a chemical module PHREEQC that considers chemistry equilibrium of solution, pure phases, gas phase, solid phase, exchanges and surfaces in Porous Media (soil and aquifer). The authors should include in the Introduction section the existence of MOHID-LAND and make comparisons. 2. There is some lack of detail on how the calculation routines for the new module are performed, namely how does it integrate salt ions reactions with the SWAT water flow and solute transport. How many parameters were used in the model calibration and validation? The data needed for SWAT modelling is not clear where it comes from, for e.g. the land cover, the soil, the crop and meteorological data (databases?). 3. For each HRU the mass of the several salt ions is generated by the several processes. In runoff how is defined the salinity percolation coefficient (ïĄć Si) and the surface runoff lag coefficient (surlag), what value is attributed and why? Explanation is needed. 4. Line 144-145, "The mass of each salt ion is routed through the channel network with water, with no chemical reactions changing in-stream salt ion concentration". Why no chemical reactions are considered in-stream to change salt ion concentration? Chemical reactions also happen in in-stream water, right? 5. Line 225, "Initial concentrations are required for each HRU." And Line 226-227 authors refer that ". . .(all HRU values are the same) concentration values yields the same result as using spatially-variable initial concentrations, if a warm-up period of several years is used in the SWAT simulation." Why it was not considered the average concentration for each sampling site spatially located near the HRU? From a theoretical point of view, does not seem correct to use as inputs non-spatially concentrations, even because the model will need a warm-up period of

several years. 6. Line 297-299, "Observed soil EC values were obtained using a saturated paste extract, and hence comparison with model results will not be as rigorous as for groundwater and surface water data." Why the comparisons with model results will not be as rigorous as for groundwater and surface water data? EC measured in a saturated paste extract (ECe) is related to the EC of the soil water (ECsw). Have you considered to use of Ayers and Westcot (1985) conversion, Skaggs et al. 2006 or using other conversion with the % saturation? 7. Line 293-294, "Only minimal manual calibration was applied to the model, to yield correct magnitudes of salt ion concentration in soil water, groundwater, and stream water." Why this approach of minimal manual calibration? And why just consider $SO_4^{2-}$ for calibration? Even understanding that from your sampling the $SO_4$ accounted for 47% of total in-stream salt mass, it would be a more solid calibration using other salt ions (especially Na), and more applicable to other studies. Can you calibrate with more salt ions? 8. Line 314, "The model does not perform as well in downstream sites, with NSE at La Junta and at Las Animas". Why the model performance is better in Rocky Ford site than in Crooked Arroyo site? What are the reasons for the weaker performance at downstream locations? Explain better in the manuscript. 9. In Fig. 14 it is observed the importance of including equilibrium chemistry into the salt transport. The no SEC simulations are underestimating the in-stream TDS. Can you explain why this underestimation is not so evident in the downstream location Las Animas? I was not expecting this.

Technical corrections: 1. All ionic forms must written considering the ionic charges (e.g. $SO_4^{2-}$, $HCO_3^-$, etc.). Correct in all the manuscript. 2. Line 59,79, 88: where its written "soil later flow" should be "soil lateral flow"? 3. Line 123: it is written "TTlag" should it be "TTlat"? 4. Line 128: where the variable Qlat,ly is described, it should refer to Qperc,ly. 5. Line 162: refer to the 8 aqueous species writing them in the ionic form. 6. Line 180: the molality is missing the subscript (mi). 7. Line 191: the equation mentions $NaCO_3^-$ that differs from the complexed specie $NaCO_3^0$ in table 1. Correction needed. 8. Line 197: there are two "in" in the sentence. 9. Line 176: C and D should be the products. 10. Line 177: Present the equation for ith 11. Line 216: It is written "(meq/100)" and

it should be "(meq/100g)". 12. Line 246: The use of commas in separation of group numbers was confusing when referring to concentrations of mg/L. In HESS guidelines for authors states that "Neither dots nor commas are permitted as group separators." Correct this in all manuscript. 13. Line 318: The sentence "Las Animas also has an $R^2$ value of 0.74." appears redundant since the $R^2$ was already commented in the previous sentence. Did the authors wanted to comment the $R^2$ for Timpas Creek? 14. Line 324: "The relationship for Crooked Arroyo yields an $R^2$ value of 0.80." This refers to data not shown? 15. Line 334: There are to "a" before stochastic in the sentence. 16. Line 382: its written "mas" and should be "mass".

---

## Editor Comment (EC1) · Christian Stamm (Editor) · 19 Apr 2019

**Editor comments**:

In addition to the comments by the reviewers, I add a few further questions:

- **L: 217**: Generally, the cation exchange capacity is *pH*-dependent. Is this taken into account by the model? If not, what are the reasons?

- **L: 293 - 295**: You mention that *only minimal manual calibration* was applied. However, changing the solubility product by almost a one order of magnitude seems more than minimal. Can you provide reasons why it may be necessary to modify a solubility product?

[Figure]

- **L: 334**: What's a stochastic river mass balance?

- **Fig. 4A**: In this figure one cannot distinguish the different ions. Please modify.

[Figure]

---

## Author Comment (AC1) · 16 May 2019

General comments: This work focused on developing a new watershed-scale salt ion fate and transport model based on SWAT model, which can account for salt loading for each major hydrologic pathway in a watershed setting for each major salt ion (SO4, Ca, Mg, Na, K, Cl, CO3, HCO3). This is very interesting work trying to quantitatively estimate the chemical and physical characteristics of the common ions, which is important for soil salinity control in semi-arid areas with shallow water table depth. Since most current research mainly focused on the transport of total salt in surface and subsurface system while not distinguish the contribution of different ions and the reactions, this work provides the new view and method for soil salinity control. I would think this work is valuable and can be published by major revision. We thank the reviewer for the

comments.

Major revisions:

(1) The numerical integrating method to couple the ion reactions and water flow and solute transport model SWAT should be illustrated in details. This will help for understanding the model.

Response: The salt_chem subroutine includes all salt chemistry reactions. The details of this subroutine have been added to Figure 2, and the following text was added to Section 2.2.7:

Lines 223-229: "The salinity chemistry reactions (precipitation-dissolution, complexation, cation exchange) are simulated for each HRU within the salt_chem subroutine (see Figure 2). Within this subroutine, the chemistry reactions are applied to the current simulated concentration values of the 5 salt minerals and the 8 salt ions for each soil layer and aquifer, to calculate new concentration values. These new concentration values are then used to simulate salt leaching (salt_lch subroutine) and salt ion loading in surface runoff (surfstor) and groundwater flow (salt_gw, substor) (Figure 2). At the end of each daily time step, the simulated salt ion mass (kg) in each transport pathway (irrigation, leaching, runoff, percolation, lateral flow, groundwater flow, dissolution/precipitation) is stored for mass balance assessment and output."

(2) How many parameters were included in this model? There is no any introduction about the parameters used in the model calibration and validation, e.g., the salinity percolation coefficient $\beta Si$, the surface runoff lag coefficient surlag. How do you set the value of these parameter, which are important to judge the reasonability of the model?

Response: The calibration and testing of the original SWAT model was presented in Wei et al. (2018). In summary, calibration was performed using SWAT-CUP. Calibration was performed for 2001-2003 using the simulated and observed streamflow at 3 stream gages in the model domain. Twenty parameters were adjusted to minimize

the objective function (see Table 4 in Wei et al., 2008). The high-sensitive parameters include SCS runoff curve number, Manning's n value for the main channel, effective hydraulic conductivity of the channel, initial volume of groundwater, recharge delay time, fraction of deep aquifer percolation, and snowfall temperature. The following text has been added to the revised manuscript:

Lines 291-298: "Calibration was performed using SWAT-CUP (Abbaspour et al., 2008) using the observed streamflow at the Rocky Ford, Las Animas, and Timpas Creek stations. Twenty parameters were targeted for modification during the calibration process, with the following exhibiting strong control on streamflow: SCS runoff curve number, Manning's n value for the main channel, effective hydraulic conductivity of the channel, initial volume of groundwater, recharge delay time, fraction of deep aquifer percolation, and snowfall temperature (Wei et al., 2018). Further details regarding calibration, model implementation, and hydrologic results are found in Wei et al. (2018)."

(3) Line 60-61," The soil water and groundwater concentration of each salt ion is also affected by equilibrium chemistry reactions: precipitation-dissolution, complexation, and cation exchange". Actually, the reactions also happen in the surface water, why not consider the chemical reactions in surface water?

Response: The reviewer has raised a valid point. However, due to the large flow and extremely high in-stream salt ion concentrations in the Arkansas River, the mass transfer of equilibrium chemistry reactions likely is negligible compared to the mass transported with advection. The application of the model to the Arkansas River Valley therefore does not depend on in-stream chemical processes. A future version of the modeling code may include in-stream equilibrium chemistry reactions.

(4) Line 294," Only minimal manual calibration was applied to the model, to yield correct magnitudes of salt ion concentration in soil water, groundwater, and stream water. Targeted parameters were the solubility product of CaSO4 precipitation-dissolution, and the soil fraction of CaSO4." Why is only the CaSO4 used to calibrate the model?

Is this due to the major ion is SO4 in this region?

Response: Correct. These two parameters / model factors were used for calibration due to the predominance of SO4 and Ca among the salt ions in the soil/groundwater system of the Arkansas River Valley. Reaction rates and fractions involving other salt ions do not have a significant effect on the total dissolved solids (TDS) in the river water. The following text has been added to clarify:

Lines 326-327: "Due to the predominance of SO4 and Ca among salt ions in the regional system, targeted parameters were the solubility product of CaSO4 precipitation-dissolution, and the soil fraction of CaSO4."

(5) What are the principle for setting the HRU with 5270? In Line 225, "Initial concentrations are required for each HRU." Were all the salt concentration of these 5270 HRU measured? Otherwise, how would you set the initial value?

Response: As discussed in Wei et al. (2018) and on Lines 266-267, each cultivated field was designated as a separate HRU. As explained on Lines 315-316, "Initial salt ion concentrations in soil water and groundwater were based on averages of observed groundwater concentrations."

Results indicate, however, that the initial concentration values for the HRUs do not have a significant effect on model results (also see Figure 14):

Lines 234-236: "However, as will be shown in Sect. 3, using uniform (i.e. all HRU values are the same) concentration values yields the same result as using spatially-variable initial concentrations, if a warm-up period of several years is used in the SWAT simulation."

(6) Line 350. The simulations for TDS and SO4 are much better than other ions, what are the possible reasons? Is this related to the targeted parameters of CaSO4 been used in calibration mentioned in Line 294? So, if the model is used in other cases, how would you choose the targeted parameters in the calibration? How about choosing

other targeted parameters in this case?

Response: Yes, the statistical measures of the simulated concentrations for SO4 in groundwater are very close to the measures of the measured concentration values, although the comparison for the other ions is also good. As mentioned by the reviewer, this may be due to the fact that the two targeted parameters (CaSO4 solubility product; soil fraction of CaSO4) have a significant control on resulting SO4 concentration values in soil water, groundwater, and river water. Results for the other ions can be improved through modifying the soil salt mineral fractions (for CaCO3, MgCO3, and NaCl). During the revision process we ran model scenarios with varying soil salt mineral fractions for these three salt minerals, and indeed the in-stream concentrations of CO3, Mg, Na, and Cl increased and were close in magnitude to the observed values. However, concentrations in the tributaries (Timpas Creek, Crooked Arroyo) were too high. We have summarized these new scenarios and results in Figure 7 and the following text:

Lines 351-361: "The cause for the under-prediction of these ions may be due to the unobserved presence of MgSO4, MgCO3, and NaCl in the soil. These minerals are not observed in NRCS soil surveys of the region, and hence were not included in the baseline model. However, several model scenarios were run to investigate the influence of these minerals. Soil bulk fractions between 0.0001 and 0.0005 were applied for these three minerals, with a large resulting effect on in-stream concentrations of Mg, Na, Cl, and CO3. For example, using a fraction of 0.0002 resulted in correct magnitude of these four ions at the Las Animas site, but over-estimated concentrations in the tributaries (e.g. Timpas Creek) (Figure 7). This model scenario, however, applied uniform salt mineral fractions of MgSO4, MgCO3, and NaCl across all 5270 HRUs. Applying spatially-varying fractions across the watershed could provide the correct magnitude of in-stream concentrations of all ions at all stream sampling sites. Regardless, measured in-stream concentrations can provide key information as to the salt minerals present in the watershed, and differences between model output and field data highlight the need for better field survey data of salt mineral content in soils."

[Figure]

(7) As shown in Fig.5, the simulation results in Rocky Ford Site are much better than those in Crooked Arroyo Site. What are the reasons? The simulation results of Na, Mg should be also shown to judge the model accuracy since the relative high concentration of these two ions as shown in Table 2.

Response: Correct: the model performs better at the Rocky Ford Site as opposed to the Crooked Arroyo site. As discussed in the text, the Rocky Ford Site is along the Arkansas River (high flows, high salt loads) whereas Crooked Arroyo is a small tributary wherein the only loadings of salt occur through non-point sources (surface runoff, lateral flow, groundwater flow, with the majority of loading via groundwater flow). As such, it is not (at least to the authors, who are familiar with the study area) surprising that model results are not as accurate as for the main stem of the river. In fact, we are quite encouraged with the level of salt loading and in-stream salt ion concentrations that were achieved by the model, as small drainage tributaries in agricultural areas are notoriously difficult to model in terms of in-stream solute concentration. This is discussed in the text:

Lines 368-371: "The relationship for Crooked Arroyo yields an $R^2$ value of 0.80. This is particularly promising given that there is no specified upstream loading for the tributaries, and hence all salt mass within the stream system is due to surface runoff, lateral flow, and groundwater discharge. Hence, comparing simulated and observed in-stream salinity concentration in these two systems is a strong test for the model."

As to the second point, Na, Mg, K, and $CO_3$ were not included in the original manuscript due to space constraints and due to the low overall contribution of these ions to the total dissolved solids concentration (particularly in the case of K and $CO_3$, which have very low concentrations in both measured data and in the model output). However, all ions have now been included in Figure 5. The Timpas Creek and Las Animas sites have also been added to Figure 5. Please also notice that the time series charts in Figure 5 (and in other figures) show only 2006-2009, the time period beyond the warm-up period and during which there are measured data. This allows the reader

to see more clearly the temporal fluctuations of the salt ion concentrations, and the comparison with the measured data.

(8) From Fig.5 and Fig.6, the simulated ion concentration fluctuated much stronger than the observed value, even the simulated value closed to zero. Is this caused by the numerical instability of coupling the ion reaction module with SWAT? Or what are the major factors resulting in the strong fluctuations?

Response: The reviewer has raised an important point. Upon further analysis, the strong fluctuations are due to the groundwater loading of salts to the river and tributaries during strong rainfall events (this can be seen by the groundwater salt loadings shown in Figure 13B, with the highest loading days coinciding with the "spikes" in the in-stream concentration plots in Figures 5 and 6). The reason for the enhanced fluctuations in the model, as compared to the measured data, is the simplistic manner in which SWAT simulates groundwater flow: with 1D steady-state flow equations rather than a physically-based, spatially-distributed method using the groundwater flow equation. This could be remedied by linking SWAT with a physically-based groundwater model such as MODFLOW, but also must include a groundwater reactive solute transport model such as RT3D.

The following text has been added to summarize this insight:

Lines 423-428: "Notice that the highest groundwater loading rates coincide with the "spikes" in the in-stream concentration plots of Figures 5 and 6, indicating the strong influence of groundwater loading on in-stream salt concentrations. The fluctuations in simulated in-stream concentration, however, are larger than observed with the measured values. This is due to the manner in which SWAT simulates groundwater return flow, with a steady-state flow equation for each HRU that provides pulses of groundwater to streams rather than the multi-dimensional groundwater flow equation that provides physically-based, spatially-distributed diffuse flow through the aquifer towards the stream network."

(9) More discussion about the contribution of different ions on salt accumulation should be added in the case discussion. Only the salt balance components for TDS were analyzed in Fig.12.

Response: We agree that a more in-depth ion-specific analysis would be helpful. However, currently the modeling code does not have the capability of outputting basin-wide salt balance information for each of the salt ions. This is due mostly to constraints on sizes of the text files, which would become inordinately large due to the detailed output for each salt ion, but also due to the fact that often a basin-wide mass balance is not performed for each salt ion, and hence the output data would not be useful. Rather, ion-specific model data are output for concentrations in soil water, groundwater, and stream water, since these values often have been measured in the field and thus are available for model testing. Later versions of the modeling code may include basin-wide mass balance components for each salt ion.

(10) Line 329-332, are the portions of salt load calculated by the model? How would you judge the reasonability of the results?

Response: Yes, the model output can be used to calculate the portions of salt load from each hydrologic pathway. Testing these values (from Figure 9) against field data is much more difficult than the direct testing/comparison of soil water concentration, groundwater concentration, and in-stream concentrations (as performed in Figures 5, 6, 7, 8, and 12). However, groundwater loadings are compared to a field estimate of mass loadings (Figure 10). Also, PERC (soil percolation) loadings are tested indirectly through the accuracy of the groundwater loadings, since groundwater salt loadings are driven in part by the amount of salt loaded to the aquifer via soil percolation.

Minor revisions:

(1) Line 33, SO4-, should be $SO_4^{2-}$. All the ions should be shown with positive and negative charges in all the other parts in the manuscript.

Response: This has been changed on Line 33 and in the Abstract, Introduction, and Methods text. However, the charges have been omitted elsewhere due to our assumption that the reader is familiar with these common ions.

(2) Line 88, "later", should be "lateral"?

Response: Yes. This has been changed.

(3) Line 133, "mas", should be "mass".

Response: This has been changed.

(4) Line 176, "C and D are reactants." Should be "C and D are products."

Response: This has been changed.

(5) Line 177, what is the equation of iA?

Response: The equation is portrayed using text: "is computed by multiplying the activity coefficient $\gamma_i$ by the molal concentration"

(6) Line 180, "mi", should be "mi".

Response: This has been changed.

(7) Line 197, there is two "in" in the sentence

Response: This has been changed.

(8) Line 250, "SO4" should be "SO42-"

Response: We have changed this to "SO4", using the common notation throughout the manuscript.

(9) Line 295, "produce" may be "product"?

Response: Yes. This has been changed.

(10)Line 382, "mas" should be "mass".

[Figure]

Response: Thank you. This has been changed.

We thank Reviewer #1 for the helpful suggestions and comments.
* * *
[Figure]

**Start**

Read Inputs **salt_read**

Required Input Data
- Initial soil / groundwater salt ion conc.
- Initial salt mineral content (%)
- Plant salt tolerance parameters

Year Loop

Day Loop

Subbasins — Water and nutrient calculations for each subbasin

HRUs — Water and nutrient calculations for each HRU in the subbasin

Rainfall/runoff hydrology
Groundwater hydrology
**Chemical equilibrium** **salt_chem**
**Irrigation loading** **salt_irrig**
Nutrient soil leaching **salt_lch**
Crop growth **soil salinity stress**
Nutrient groundwater transport **salt_gw**
Lag nutrients **and salt** in surface runoff **surfstor**
Lag nutrients **and salt** in groundwater flow **substor**

**For each soil layer and aquifer of HRU** *j*

Calculate water content

Calculate ionic strength

Calculate activity coefficient

Compute prec.-dissolution

Compute cation exchange

Salt mass dissolved/prec.

Route Water — Route water, sediment, nutrients, **and salt** **watqual** through the stream network

Mass calculations **salt_balance**

- salt.output.std
- salt.output.rch
- salt.output.sub
- salt.output.hru

Output Data

**End**

**Fig. 1.** Figure 2
Interactive
comment

A **Las Animas**
(main stem)

B **Timpas Creek**
(tributary)

**Fig. 2.** Figure 7
Interactive
comment

**Fig. 3.** Figure 5

Rocky Ford | A

NSE = 0.68

Timpas Creek | B

NSE = -0.29

Crooked Arroyo | C

NSE = 0.65

La Junta | D

NSE = 0.26

Las Animas | E

NSE = 0.19

**Fig. 4.** Figure 6

---

## Author Comment (AC2) · 16 May 2019

General comments: This work aims at simulating the fate and transport of 8 major salt ions ($SO_4^{2-}$, $Ca^{2+}$, $Mg^{2+}$, $Na^+$, $K^+$, $Cl^-$, $CO_3^{2-}$, $HCO_3^-$) in a watershed hydrologic system using a new salinity transport module implemented in the SWAT code. This modelling code for salt transport includes surface runoff, percolation, soil lateral flow, groundwater flow and streamflow and also considers equilibrium chemistry reactions in soil layers and aquifers. This paper addresses with an interesting and practical approach the concerning thematic of soil and aquifer salinization. This study uses a quantification approach with salt balances performed in the watershed, includes the constituent mass in irrigation water, and the contribution of each salt ion to the salinity, which is less seen in published studies were the focus is the total of salts. Also, considering the new tool proposed that helps in predicting the impact of irrigation practices and in controlling salinity, I suggest the publication of this work after major revision.

We thank the reviewer for the comments.

Specific comments:

1. Line 53 "Currently, there is no model that simulates salt trans-port in all major hydrologic pathways (surface runoff, soil percolation and leaching, groundwater flow, streamflow) at the watershed-scale that also considers important solution reaction chemistry." Actually there is MOHID LAND model that is also cou-pled with SWAT. MOHID LAND is a physically-based, spatially distributed, continuous, variable time step model for the water and property cycles in inland waters and main mediums that also includes a chemical module PHREEQC that considers chemistry equilibrium of solution, pure phases, gas phase, solid phase, exchanges and surfaces in Porous Media (soil and aquifer). The authors should include in the Introduction section the existence of MOHID-LAND and make comparisons.

Response: Thank you for the information. We were not aware of this model. However, we are not able to find any publications that describe the PHREEQC module for the MOHID modeling framework – we are only able to find a few references in on-line posts and a link to the source code. Also, the only reference to the linkage between SWAT and MOHID that we can find is a conference paper ("Integration of MOHID Model and Tools with SWAT Model", from a 2007 SWAT conference). We would be happy to include a description of the linkage between SWAT and MOHID-PHREEQC, if the reviewer can provide references to published journal articles.

2. There is some lack of detail on how the calculation routines for the new module are performed, namely how does it integrate salt ions reactions with the SWAT water flow and solute transport. How many parameters were used in the model calibration and validation? The data needed for SWAT modelling is not clear where it comes from, for e.g. the land cover, the soil, the crop and meteorological data (databases?).

Response: We have added text to describe each of these points:

Line 223-229: "The salinity chemistry reactions (precipitation-dissolution, complexation, cation exchange) are simulated for each HRU within the salt_chem subroutine (see Figure 2). Within the salt_chem subroutine, the chemistry reactions are applied to the current simulated concentration values of the 5 salt minerals and the 8 salt ions for each soil layer and aquifer, to calculate new concentration values. These new concentration values are then used to simulate salt leaching (salt_lch subroutine) and salt ion loading in surface runoff (surfstor) and groundwater flow (salt_gw, substor) (Figure 2). At the end of each daily time step, the simulated salt ion mass (kg) in each transport pathway (irrigation, leaching, runoff, percolation, lateral flow, groundwater flow, dissolution/precipitation) is stored for mass balance assessment and output."

Lines 282-283: "The digital elevation model (DEM), stream network, soil map, land-use map, climate data, streamflow, and canal diversion data were obtained from the USGS, NRCS, and several state agencies, as summarized in Wei et al. (2018)."

Lines 293-298: "Calibration was performed using SWAT-CUP (Abbaspour et al., 2008) using the observed streamflow at the Rocky Ford, Las Animas, and Timpas Creek stations. Twenty parameters were targeted for modification during the calibration process, with the following exhibiting strong control on streamflow: SCS runoff curve number, Manning's n value for the main channel, effective hydraulic conductivity of the channel, initial volume of groundwater, recharge delay time, fraction of deep aquifer percolation, and snowfall temperature (Wei et al., 2018). Further details regarding calibration, model implementation, and hydrologic results are found in Wei et al. (2018)."

3. For each HRU the mass of the several salt ions is generated by the several processes. In runoff how is defined the salinity percolation coefficient (ï ÌÍA ÌÀc Si) and the surface runoff lag coefficient (surlag), what value is attributed and why? Explanation is needed.

Response: The concentration of salinity in surface runoff is determined by the salinity

percolation coefficient (0 to 1), which in this model is assumed to be the same as the nitrate percolation coefficient ( = 0.2). Therefore, surface runoff salinity concentration is 20% of the concentration value of the salinity in percolate water. The surface runoff lag coefficient is 2.0 days, and was not adjusted for any HRU during calibration of the salinity model. This value was determined during the calibration of the hydrologic model in Wei et al. (2018).

4. Line 144-145,"The mass of each salt ion is routed through the channel network with water, with no chemical reactions changing in-stream salt ion concentration". Why no chemical reactions are considered in-stream to change salt ion concentration? Chemical reactions also happen in in-stream water, right?

Response: The reviewer has raised a valid point. However, due to the large flow and extremely high in-stream salt ion concentrations in the Arkansas River, the mass transfer of equilibrium chemistry reactions likely is negligible compared to the mass transported with advection. The application of the model to the Arkansas River Valley therefore does not depend on in-stream chemical processes. A future version of the modeling code may include in-stream equilibrium chemistry reactions.

5. Line 225, "Initial concentrations are required for each HRU." And Line 226-227 authors refer that "...(all HRU values are the same) concentration values yields the same result as using spatially-variable initial concentrations, if a warm-up period of several years is used in the SWAT simulation." Why it was not considered the average concentration for each sampling site spatially located near the HRU? From a theoretical point of view, does not seem correct to use as inputs non-spatially concentrations, even because the model will need a warm-up period of several years.

Response: Certainly spatially-dependent values of soil and groundwater salt ion concentrations can be used to initialize HRU values, and therefore be more accurate at the onset of the model simulation period. We assumed that this would be necessary. However, during scenario testing it was observed that the model results, at least for

this study region, are not significantly sensitive to initial conditions, given several years of warm-up. This is the point of the scenario and associated conclusion, which is presented in Section 3.3.2.4 ("Scenarios and Model Guidelines"):

Lines 440-442: "There are only small differences between using uniform or HRU-variable initial concentrations for soil water and groundwater. Any differences are readily resolved during the warm-up period. Hence, to facilitate model use we recommend that uniform initial concentrations be used."

6. Line 297-299, "Observed soil EC values were obtained using a saturated paste extract, and hence comparison with model results will not be as rigorous as for groundwater and surface water data." Why the comparisons with model results will not be as rigorous as for groundwater and surface water data? EC measured in a saturated paste extract (ECe) is related to the EC of the soil water (ECsw). Have you considered to use of Ayers and Westcot (1985) conversion, Skaggs et al. 2006 or using other conversion with the % saturation?

Response: Thank you for commenting on this. We agree that we should compare estimated field-measured EC of soil paste extract with estimated simulated values. This was performed during the revision process by converting soil water TDS to ECw, and then to ECe using the ratio of soil water (mm) to water amount at saturation (mm) for the SWAT soil layers. This was performed for all cultivated HRUs during the 2002-2005 growing season, coinciding with the period of field sampling. The SWAT code was modified to output these data. The results are shown in Figure 12D (revised manuscript) using frequency distributions of the observed and simulated values. The following text was added to describe the field surveys and then provide analysis of results:

Lines 272-278: "Average soil water salinity, based on electrical conductivity of a soil paste extract (ECe), is 4.11 dS/m (54700 measurements), with minimum and maximum of 0.9 dS/m and 56.5 dS/m, respectively (Morway and Gates, 2012). These values

were estimated from measurements of apparent bulk soil conductivity, taken with a Geonics EM-38 electromagnetic induction sensor, as described in Morway and Gates (2012). Surveys were performed during the months of March-September for 1999-2005. Based on 6 surface water sampling sites (4 in the Arkansas River, 2 in tributaries; Figure 3B), average and is 1145 mg/L and 560 mg/L, respectively. More details of observed groundwater, soil water, and surface water concentrations are provided in Sect. 3.3.2 when model results are presented."

Lines 402-411: "A relative frequency plot of observed and simulated ECe (dS/m) in the soil profile is shown in Figure 12D. The simulated values were taken from HRUs coinciding with cultivated fields for the days of April 15, May 15, June 15, July 15, and August 15, for the years 2001-2005. Note that simulated values were taken from each cultivated HRU, whereas the field surveys using the EM-38 sensors were conducted in approximately 100 fields. The average of observed values is 4.1 dS/m, although this number is skewed by extremely high values (> 30 dS/m). If only values < 6.5 dS/m are considered (89% of the samples), then the average is 3.2 dS/m. The average of the simulated values is 2.96 dS/m. As seen from the frequency distribution in Figure 12D, the model tends to under-estimate soil salinity for some of the HRUs, and does not capture the high salinity values (> 7 dS/m). However, the overall magnitude and distribution of values approaches the distribution of the measured values. Note that EM-38 measurements have inherent uncertainty. In addition, some of the HRUs in-cluded in the analysis are fallow during this period (2002-2005), which may lead to low soil salinity values that were not measured in the field survey."

7. Line 293-294, "Only minimal manual calibration was applied to the model, to yield correct magnitudes of salt ion concentration in soil water, groundwater, and stream water." Why this approach of minimal manual calibration? And why just consider $SO_4^{2-}$ for calibration? Even understanding that from your sampling the $SO_4$ accounted for 47% of total in-stream salt mass, it would be a more solid calibration using other salt ions (especially Na), and more applicable to other studies. Can you calibrate with more

salt ions?

Response: The word "minimal" was used to indicate that only two parameters were varied during model calibration. We changed the wording to read:

Lines 325-327: "Manual calibration was applied to the model to yield correct magnitudes of salt ion concentration in soil water, groundwater, and stream water. Due to the predominance of $SO_4$ and $Ca$ among salt ions in the regional system, targeted parameters were the solubility product of $CaSO_4$ precipitation-dissolution and the soil fraction of $CaSO_4$."

However, to the reviewer's point, parameters governing the other salt minerals ($CaCO_3$, $MgCO_3$, and $MgSO_4$) could be varied to provide a better match between observed and simulated salt ion concentrations in the groundwater and river water. We tested this during the revision process, running model scenarios with varying soil fractions of these three salt minerals. Indeed, the in-stream concentrations of $CO_3$, $Mg$, $Na$, and $Cl$ increased and were close in magnitude to the observed values. However, concentrations in the tributaries (Timpas Creek, Crooked Arroyo) were too high. Therefore, perhaps unobserved fractions of these salt minerals may be present in the watershed soils. We have summarized these new scenarios and results in Figure 7 and the following text:

Lines 351-361: "The cause for the under-prediction of these ions may be due to the unobserved presence of $MgSO_4$, $MgCO_3$, and $NaCl$ in the soil. These minerals are not observed in NRCS soil surveys of the region, and hence were not included in the baseline model. However, several model scenarios were run to investigate the influence of these minerals. Soil bulk fractions between 0.0001 and 0.0005 were applied for these three minerals, with a large resulting effect on in-stream concentrations of $Mg$, $Na$, $Cl$, and $CO_3$. For example, using a fraction of 0.0002 resulted in correct magnitude of these four ions at the Las Animas site, but over-estimated concentrations in the tributaries (e.g. Timpas Creek) (Figure 7). This model scenario, however, applied uniform

salt mineral fractions of MgSO4, MgCO3, and NaCl across all 5270 HRUs. Applying spatially-varying fractions across the watershed could provide the correct magnitude of in-stream concentrations of all ions at all stream sampling sites. Regardless, measured in-stream concentrations can provide key information as to the salt minerals present in the watershed, and differences between model output and field data highlight the need for better field survey data of salt mineral content in soils."

8. Line 314, "The model does not perform as well in downstream sites, with NSE at La Junta and at Las Animas". Why the model performance is better in Rocky Ford site than in Crooked Arroyo site? What are the reasons for the weaker performance at downstream locations? Explain better in the manuscript.

Response: Likely, the model performs better at the Rocky Ford site due to the proximity to the upstream end of the watershed, where loading for each salt ion is specified for each day. However, through visual inspection (Figure 6), the model performs adequately in simulating the temporal fluctuation and magnitude of TDS at the La Junta gage, with only one measured concentration value, from January 17, 2009, much different than the simulated value – this is actually due to an over-estimation of streamflow by SWAT, and thereby an under-prediction of in-river concentration.

However, during the revision process we noticed that we were using an old version of the SWAT model, which over-estimated flow in the downstream reaches of the watershed, and thus under-estimate the in-stream salt ion concentrations. Using the most up-to-date version of the model (as seen in Wei et al., 2018), the downstream flows match the observed flows much more closely, and hence the simulated in-stream salt ion concentrations are much closer in magnitude to the measured values. This can be seen in Figure 5D and Figure 6E for the Las Animas site.

9. In Fig. 14 it is observed the importance of including equilibrium chemistry into the salt transport. The no SEC simulations are underestimating the in-stream TDS. Can you explain why this underestimation is not so evident in the downstream location Las

[Figure]

Animas? I was not expecting this.

Response: This effect at Las Animas was due to the use of the outdated SWAT model, which overestimated flow in the downstream reaches of the Arkansas River (see response to previous comment). Using the up-to-date SWAT model, the results for the Las Animas site (Figure 15C) (i.e. under-predicting in the scenario of no SEC) are similar to other sites. However, notice that the results for the Rocky Ford site (Figure 15A) show only small differences between the scenarios. For the Rocky Ford site, the scenarios yield similar results due to the location of the site being close to the upstream end of the modeled region, and thus in-stream concentrations are not affected by groundwater and surface runoff salt loadings to the river (Lines 464-466).

Technical corrections:

1. All ionic forms must written considering the ionic charges (e.g.$SO_4^{2-}$, $HCO_3^-$, etc.). Correct in all the manuscript. Response: The charges are included in Table 1 and in the Introduction and Methods text, but omitted elsewhere due to our assumption that the reader is familiar with these common ions.

2. Line 59,79, 88: where its written "soil later flow" should be "soil lateral flow"? Response: This has been changed.

3. Line 123: it is written "TTlag" should it be "TTlat"? Response: Yes. This has been changed.

4. Line 128: where the variable $Q_{lat,ly}$ is described, it should refer to $Q_{perc,ly}$. Response: Thank you. This has been changed.

5. Line 162: refer to the 8 aqueous species writing them in the ionic form. Response: This has been changed.

6. Line 180: the molality is missing the subscript ($m_i$). Response: This has been changed.

7. Line 191: the equation mentions NaCO3-that differs from the complexed specie NaCO30 in table 1. Correction needed. Response: This has been corrected.

8. Line197: there are two "in" in the sentence. Response: This has been changed.

9. Line 176: C and D should be the products. Response: This has been changed.

10. Line 177: Present the equation for ith Response: This is provided using text.

11. Line 216: It is written "(meq/100)" and it should be "(meq/100g)". Response: This has been changed.

12. Line 246: The use of commas in separation of group numbers was confusing when referring to concentrations of mg/L. In HESS guidelines for authors states that "Neither dots nor commas are permitted as group separators." Correct this in all manuscript. Response: Thank you. Commas have been removed from numbers throughout the manuscript.

13. Line 318: The sentence "Las Animas also has an R2 value of 0.74." appears redundant since the R2 was already commented in the previous sentence. Did the authors wanted to comment the R2 for Timpas Creek? Response: Yes. This has been changed.

14.Line 324: "The relationship for Crooked Arroyo yields an R2 value of 0.80." This refers to data not shown? Response: Yes. This has been changed in the text.

15. Line 334: There are to "a" before stochastic in the sentence. Response: Thank you. This has been corrected.

16. Line 382: its written "mas" and should be "mass". Response: This has been changed.

We thank Reviewer #2 for the helpful suggestions and comments.
* * *
614, 2019.

**Start**

Read Inputs   **salt_read**

Required Input Data
- Initial soil / groundwater salt ion conc.
- Initial salt mineral content (%)
- Plant salt tolerance parameters

Year Loop

Day Loop

Subbasins   Water and nutrient calculations for each subbasin

HRUs   Water and nutrient calculations for each HRU in the subbasin

**For each soil layer and aquifer of HRU _j_**

Rainfall/runoff hydrology
Groundwater hydrology
**Chemical equilibrium**   **salt_chem**
**Irrigation loading**   **salt_irrig**
Nutrient soil leaching   **salt_lch**
Crop growth   **soil salinity stress**
Nutrient groundwater transport   **salt_gw**
Lag nutrients **and salt** in surface runoff   *surfstor*
Lag nutrients **and salt** in groundwater flow   *substor*

Calculate water content

Calculate ionic strength

Calculate activity coefficient

Compute prec.-dissolution

Compute cation exchange

Salt mass dissolved/prec.

Route Water   Route water, sediment, nutrients, **and salt** through the stream network   *watqual*

Mass calculations   **salt_balance**

Output Data
- salt.output.std
- salt.output.rch
- salt.output.sub
- salt.output.hru

**End**

**Fig. 1.** Figure 2

[Figure]

**Fig. 2.** Figure 5

**Fig. 3.** Figure 6

[Figure]

A   **Las Animas**
     (main stem)

B   **Timpas Creek**
     (tributary)

**Fig. 4.** Figure 7

[Figure]

**Fig. 5.** Figure 12

[Figure]

[Figure]

**Fig. 6.** Figure 15

---

## Author Comment (AC3) · 16 May 2019

L: 217: Generally, the cation exchange capacity is pH-dependent. Is this taken into account by the model? If not, what are the reasons?

Response: pH was not simulated in the model. The salinity module used in SWAT-Salt is based on Tavakoli-Kivi et al. (2019: "A salinity reactive transport and equilibrium chemistry model for regional-scale agricultural groundwater systems. Journal of Hydrology 572, 274-293"), which does not account for pH. The module was not changed in this sense for imbedding within SWAT. In addition, the precipitation-dissolution reactions dwarf the cation exchange process in terms of governing salt ion concentration, and hence we believe that the exclusion of pH dependency is not critical for this study

region. It will be re-visited for future studies and model applications.

L: 293 - 295: You mention that only minimal manual calibration was applied. However, changing the solubility product by almost a one order of magnitude seems more than minimal. Can you provide reasons why it may be necessary to modify a solubility product?

Response: The word "minimal" in the text refers to the low number (2) of parameters modified during manual calibration. This has been changed in the text:

Lines 325-327: "Manual calibration was applied to the model to yield correct magnitudes of salt ion concentration in soil water, groundwater, and stream water. Due to the predominance of $SO_4$ and $Ca$ among salt ions in the regional system, targeted parameters were the solubility product of $CaSO_4$ precipitation-dissolution and the soil fraction of $CaSO_4$."

Similar to groundwater salinity models that employ equilibrium chemistry, simulations indicate that model results are strongly dependent on the solubility product of the salt minerals. These solubility products are governed principally by temperature and pH. As temperature in the soil profile and aquifer differ, and also vary seasonally, and since pH is not modeled in the current model version, the solubility product of $CaSO_4$ was modified during the calibration process since the true solubility product value is not known with certainty. However, the same value was used for both the soil profile and aquifer, with the value held constant for all HRUs.

L: 334: What's a stochastic river mass balance?

Response: This refers to a salinity mass balance of the Arkansas River system, . For clarity, we have changed the text to the following:

Lines 381-384: "Mass balance plot values are the mean of an ensemble of a stochastic river mass balance calculation of surface water salinity loadings along the length of the Arkansas River within the model domain, using a method similar to Mueller-Price and

[Figure]

Gates (2008), with values indicating the mass of salt not accounted for by surface water loadings.

Fig. 4A: In this figure one cannot distinguish the different ions. Please modify.

Response: This figure was changed to show average daily salt ion loading, for each year (1999-2009). The values for each salt ion can now be seen more clearly.

We thank the Editor for the helpful suggestions and comments.

—————————————————————

[Figure]

[Figure]

**Fig. 1.** Figure 4